# Training-Free Determination of Network Width via Neural Tangent Kernel

**Tatsumi Sunada**[1,2]   **Toshihiko Yamasaki**[1]   **Atsuto Maki**[2]
[1]The University of Tokyo    [2]KTH Royal Institute of Technology
{sunada,yamasaki}@cvm.t.u-tokyo.ac.jp, atsuto@kth.se

## Abstract

Determining an appropriate size for an artificial neural network under computational constraints is a fundamental challenge. This paper introduces a practical metric, derived from Neural Tangent Kernel (NTK), for estimating the minimum necessary network width with respect to test loss *prior to training*. We provide both theoretical and empirical evidence that the smallest eigenvalue of the NTK strongly influences test loss in wide but finite-width neural networks. Based on this observation, we define an NTK-based metric computed at initialization to identify what we call *cardinal width*, i.e., the width of a network at which generalization performance saturates. Our experiments across multiple datasets and architectures demonstrate the effectiveness of this metric in estimating the *cardinal width*.

## 1 Introduction

Selecting an appropriate width for modern deep networks is a crucial challenge, especially under computational constraints. In overparameterized regimes, models tend to reduce generalization loss as the width increases (Belkin et al., 2019; Nakkiran et al., 2021). In practice, however, the marginal improvement saturates once the width becomes sufficiently large. This motivates the need for a principled criterion to determine the point where additional width is no longer useful.

Beyond simple cross-validation for width selection, prior work has explored three main directions: adopting architectures to tasks and computational resources while training the models (Liu et al., 2017; Gordon et al., 2018; He et al., 2018; Yang et al., 2018; Yu & Huang, 2019; Yu et al., 2019; Cai et al., 2020), neural architecture search (NAS) without training (Abdelfattah et al., 2021; Mellor et al., 2021), and scaling laws for the size of models (Kaplan et al., 2020; Hoffmann et al., 2022). Although these methods reduce search effort, they largely lack a theory-grounded criterion for width determination. Width selection remains ad hoc in the absence of such a theory. These methods can rank or constrain architectures, but they still lead to repeated trial-and-error without a clear rule indicating the point at which increasing width stops improving performance.

To address this issue, we introduce a *training-free* method for width determination based on the smallest eigenvalue of the Neural Tangent Kernel (NTK), derived from our theoretical analysis. Specifically, we show that in the infinite-width regime, where gradient-based training with squared loss is equivalent to kernel ridgeless regression via NTK linearization (Jacot et al., 2018; Lee et al., 2019), an upper bound on the test error is governed by the smallest eigenvalue of the NTK. In addition, for a sufficiently large but finite width, we show that an analogous bound holds with the empirical NTK. We then demonstrate the validity of the theory with an experiment on two-layer networks. These theoretical and empirical results suggest that one can find the minimum sufficient width by monitoring the smallest NTK eigenvalue across widths and when its growth saturates. Based on these findings, we propose a method for using the smallest eigenvalue of the NTK to identify the width at which generalization performance saturates, which we call *cardinal width*. Our experimental results show that the NTK-based method of determining appropriate width is effective across multiple architectures and datasets. Our contributions can be summarized as follows:

1. **Infinite-width Theory:** We show that the test error of infinite-width networks (which is analyzed in the framework of kernel ridgeless regression) admits an upper bound proportional to $\mu_{\min}^{-2}$, where $\mu_{\min}$ denotes the smallest eigenvalue of the NTK.

2. **Finite-width Theory:** We show that for sufficiently wide but finite-width networks, the upper bound of the test error is controlled by $\mu_{\min}^{-2}$.

3. **Training-free width selection:** We propose a method to determine the *cardinal width* based on $\mu_{\min}$ that provides a principled pre-training stopping point for increasing width.

4. **Experiments:** Across multiple architectures and datasets, the predicted saturation width aligns with the *cardinal width*, indicating the effectiveness of the proposed method.

## 2 RELATION TO PRIOR WORKS

**Width selection.**  Studies on width determination can be divided into three categories. Many studies address width selection through supernet training (Liu et al., 2017; Gordon et al., 2018; He et al., 2018; Yang et al., 2018; Yu & Huang, 2019; Cai et al., 2020). These methods change the model size during training or train multiple models of different sizes in parallel. Another line of research employs a training-free metric to rank architectures to design a network before training (Chen et al., 2021; Mellor et al., 2021; Xu et al., 2021). Although Chen et al. (2021) utilize a theoretical tool (NTK) for the metric, they do not provide any explicit theory that connects NTK to the model's performance. Recently, scaling laws for models have been proposed (Tan & Le, 2019; Kaplan et al., 2020; Hoffmann et al., 2022), which indicate how to scale the model size along with changing data size. These methods substantially reduce search costs but are generally based on empirical evidence and lack a theory-backed stopping rule that certifies when further widening ceases to improve generalization. Our work provides a clear rule for width selection via an NTK-based theoretical analysis.

**Generalization of kernel ridge (less) regression (KRR).**  KRR theory characterizes excess risk through Mercer–eigenvalue decay and effective dimension (Caponnetto & De Vito, 2007; Steinwart et al., 2009). More recent analyses provide precise risk characterizations in terms of the full kernel eigenspectrum, including closed-form spectral decompositions and filters (Canatar et al., 2021; Simon et al., 2023; Cheng et al., 2024). These works leverage the entire spectrum of a kernel matrix to obtain tight bounds. In contrast, we show that an upper bound on the test error can be controlled by the *single* smallest eigenvalue so that the width selection is computationally efficient compared to using full eigenvalues. Since infinite-width neural networks coincide with kernel regression with NTK (Jacot et al., 2018; Lee et al., 2019), our results of KRR directly extend to the setting in infinite-width neural networks.

**Neural Tangent Kernel (NTK).**  The NTK connects gradient descent in very wide networks to kernel regression (Jacot et al., 2018; Lee et al., 2019). Classical results rely on the *infinite-width* or *lazy-training* idealization, in which features remain close to initialization (Chizat et al., 2019). This yields an elegant theory but overlooks effects that arise at realistic widths. To address this gap, recent work has extended NTK analysis to finite-width regimes, including finite-width corrections (Hanin & Nica, 2020; Huang & Yau, 2020), concentration results (Lee et al., 2019), and fluctuation analyses (Bordelon & Pehlevan, 2023). Within this literature, several studies have highlighted the importance of the smallest eigenvalue of the NTK, since it governs kernel conditioning and thus affects both optimization and generalization (Nguyen et al., 2021; Montanari & Zhong, 2022; Karhadkar et al., 2024). Although the smallest eigenvalue is widely known as an important factor for generalization thanks to these works, the prior work stops short of directly connecting the smallest eigenvalue to generalization error. To the best of our knowledge, our paper is the first to establish such a connection, showing that the generalization behavior of both infinite and finite-width networks can be characterized via the smallest eigenvalue.

**Summary of differences from related works**

- Unlike many of prior approaches to width selection that are empirical or limited in theory, we present a training-free criterion that is grounded in NTK theory. It selects the cardinal width by comparing changes in the smallest eigenvalue of the NTK across widths.

- We analyze kernel ridgeless regression through the single smallest eigenvalue of the NTK, whereas prior bounds typically rely on the full spectrum. The smallest eigenvalue alone controls an upper bound on test error, enabling practical estimation via minimal eigenvalue computation instead of full eigendecompositions.

- To our knowledge, this is the first work that links the smallest eigenvalue of the empirical NTK at finite width directly to an upper bound of the generalization error. Prior works on the smallest eigenvalue mainly established the positive definiteness of NTK or bounded the eigenvalue itself, while our analysis connects it to test performance.

# 3 THEORETICAL ANALYSIS

## 3.1 PROBLEM SETUP

In this study, we begin our theoretical analysis by exploring the relationship between the smallest eigenvalue of the NTK and test error in infinite-width networks, analyzed within the framework of kernel ridgeless regression (Jacot et al., 2018; Lee et al., 2019). Then, we extend this analysis to *finite*-width networks trained by gradient flow. In this section, we formally define the setting of kernel ridge (less) regression, NTK, and *finite*-width neural networks by gradient flow.

**Kernel ridge (less) regression.** Let $\mathcal{D}$ be an unknown distribution over $\mathcal{X} \times \mathcal{Y} \subseteq \mathbb{R}^d \times \mathbb{R}$ and let $\{(\boldsymbol{x}_i, y_i)\}_{i=1}^n \sim \mathcal{D}$ be a dataset of size $n$. Assume the outputs follow a target function $f^* : \mathcal{X} \to \mathbb{R}$ with noise $\epsilon_i \sim \mathcal{N}(0, \sigma^2)$, i.e., $y_i = f^*(\boldsymbol{x}_i) + \epsilon_i$. Given a ridge parameter $\alpha \geq 0$, Kernel Ridge Regression (KRR) solves

$$\hat{f} = \underset{f \in \mathcal{H}}{\operatorname{argmin}} \frac{1}{n} \sum_{i=1}^n \left( f(\boldsymbol{x}_i) - y_i \right)^2 + \alpha \|f\|_{\mathcal{H}}^2, \tag{1}$$

where $\mathcal{H}$ is the Reproducing Kernel Hilbert Space (RKHS) induced by a symmetric positive (semi) definite kernel $K : \mathcal{X} \times \mathcal{X} \to \mathbb{R}$. Since the infinite-width network in the NTK-regime is analyzed in the framework of kernel *ridgeless* regression, we specifically focus on the case $\alpha \to 0$.

Let $X := \{\boldsymbol{x}_i\}_{i=1}^n$ and $\boldsymbol{y} := (y_1, \ldots, y_n)^\top$. Given this setup, KRR predicts a function $\hat{f}$ as

$$\hat{f}(\boldsymbol{x}) = \boldsymbol{k}(\boldsymbol{x}, X) \left( K + n\alpha \, \boldsymbol{I}_n \right)^{-1} \boldsymbol{y}, \tag{2}$$

where $K$ denotes the kernel matrix with entries $K(\boldsymbol{x}_i, \boldsymbol{x}_j)$, $\boldsymbol{k}(\boldsymbol{x}, X)$ is the row vector with entries $K(\boldsymbol{x}, \boldsymbol{x}_i)$, and $\boldsymbol{I}_n$ denotes the $n \times n$ identity matrix.

**Neural Tangent Kernel.** An infinite-width neural network is regarded as the function $\hat{f}$ in Equation 2, where $K$ is replaced by NTK. Given the dataset and a functional output $f_{\boldsymbol{\theta}}(\boldsymbol{x})$ of a neural network with parameters $\boldsymbol{\theta}$ for input $\boldsymbol{x}$, the NTK matrix $K_\infty \in \mathbb{R}^{n \times n}$ is defined as

$$\left( K_\infty \right)_{ij} = \mathbb{E}_{\boldsymbol{\theta}_0} \left[ \nabla_{\boldsymbol{\theta}} f_{\boldsymbol{\theta}_0}(\boldsymbol{x}_i)^\top \nabla_{\boldsymbol{\theta}} f_{\boldsymbol{\theta}_0}(\boldsymbol{x}_j) \right], \tag{3}$$

where the expectation is taken over the random initialization of $\boldsymbol{\theta}_0$.

**Finite-width neural network.** Here, we describe the training process of an $m$-width neural network $f_m$ on a regression problem. The network is trained by gradient flow while minimizing the scaled mean squared error (MSE) loss $\mathcal{L} := \frac{1}{2n} \sum_{i=1}^n \left( f_m(\boldsymbol{x}_i; \boldsymbol{\theta}) - y_i \right)^2$. The network parameters are updated by gradient flow $\frac{d\boldsymbol{\theta}}{dt} = -\eta \frac{\partial \mathcal{L}}{\partial \boldsymbol{\theta}}$, where $\eta$ is the learning rate and $t \geq 0$ is the training time. Let $f_m(\cdot, t)$ denote the network at time $t$, and we consider the terminal time $T$, producing $f_m(\cdot, T)$. We also define $K_m^{(t)}$ as the empirical NTK matrix at time $t$ on the training inputs, computed as in Equation 3 but without the initialization expectation, i.e.,

$$\left( K_m^{(t)} \right)_{ij} = \nabla_{\boldsymbol{\theta}} f_{\boldsymbol{\theta}(t)}(\boldsymbol{x}_i)^\top \nabla_{\boldsymbol{\theta}} f_{\boldsymbol{\theta}(t)}(\boldsymbol{x}_j). \tag{4}$$

## 3.2 MAIN RESULTS

We now present the key theorems of this paper that establish the relationship between the smallest eigenvalue of the NTK and the generalization error in both *infinite*- and *finite*-width networks. We begin with the *infinite*-width case via the framework of kernel ridgeless regression.

**Infinite-width case.** We consider kernel ridgeless regression introduced in Section 3.1. Given a dataset $\{(\boldsymbol{x}_i, y_i)\}_{i=1}^n \sim \mathcal{D}$, let $X$ be the training inputs, $K : \mathcal{X} \times \mathcal{X} \to \mathbb{R}$ be a positive definite kernel and $G := K(X, X) \in \mathbb{R}^{n \times n}$ its Gram matrix with entries $G_{ij} = K(\boldsymbol{x}_i, \boldsymbol{x}_j)$. Let the eigenvalues of the matrix be ordered as $\mu_1 \geq \mu_2 \geq \cdots \geq \mu_n \geq 0$, and denote its smallest eigenvalue $\mu_n$ by $\mu_{\min}$. Let $\hat{f}$ be the predictor obtained by kernel ridgeless least squares with kernel $K$ on $X$. We measure generalization error as

$$E_g := \mathbb{E}_{(X, \boldsymbol{y}) \sim \mathcal{D}} \big[ (\hat{f}(\boldsymbol{x}) - f^*(\boldsymbol{x}))^2 \big]. \tag{5}$$

In the infinite-width NTK regime, gradient flow on neural networks with squared loss corresponds to kernel ridgeless regression under NTK (Jacot et al., 2018; Lee et al., 2019). Thus, the following results apply to infinite-width networks by taking $K$ to be the NTK.

**Assumption 3.1** (Exclusion of the interpolation peak). We exclude problem instances that are arbitrarily close to the double-descent peak. In the closed-form risk expression by Canatar et al. (2021) that we use later, a scalar factor $\gamma \in [0, 1)$ quantifies the closeness of a model to the interpolation peak. Here, we assume $1 - \gamma \geq c_0$ for some universal constant $c_0 > 0$. Intuitively, the infinite-width model has far more effective parameters than the size of data. The peak occurs at the point where these numbers are equal, supporting the validity of this assumption.

**Theorem 3.2** (Infinite-width error bound via the smallest eigenvalue). *Under Assumption 3.1, there exist absolute constants $C_1, C_2 > 0$ independent of $\mu_{\min}$ and the width, such that the test error $E_g$ satisfies*

$$E_g \leq C_1 \mu_{\min}^{-2} + C_2 \sigma^2 n \mu_{\min}^{-2}. \tag{6}$$

This bound shows that the *single* spectral quantity $\mu_{\min}$ controls an upper bound on the test error. For a fixed dataset, both $n$ and $\sigma$ can be regarded as constants, so the bound simplifies to $E_g \leq C \mu_{\min}^{-2}$ for an absolute constant $C$. Practically, estimating only $\mu_{\min}$ is far cheaper than computing the full eigenspectrum. Thus, this simple upper bound with $\mu_{\min}$ provides a natural starting point for efficiently determining an appropriate width. We emphasize that the bound characterizes worst-case behavior. In practice, the empirical relationship between $\mu_{\min}$ and test error can be substantially milder, as we demonstrate in Section 3.4.

**Finite-width case.** Theorem 3.2 holds on an infinite-width network, which does not apply in practice. We now pass from the infinite-width NTK predictor to *finite-width* neural networks trained by gradient flow, following the setup in Sec. 3.1. For a network of width $m$ evaluated at terminal time $T$, we measure generalization by

$$E_g^{(m)} := \mathbb{E}_{(X, \boldsymbol{y}) \sim \mathcal{D}} \big[ (f(\boldsymbol{x}) - f_m(\boldsymbol{x}, T))^2 \big]. \tag{7}$$

We rewrite $E_g^{(\infty)}$ for the test error of the infinite-width (NTK) regressor characterized in Theorem 3.2. Let $K_m^{(u)}$ be the empirical NTK at time $u \in [0, T]$ for width $m$, and let $K_\infty$ be the infinite-width NTK. For any kernel matrix $K$, let $\mu_{\min}(K)$ denote its smallest eigenvalue. In the following, we relate $E_g^{(m)}$ to $E_g^{(\infty)}$ via the minimal eigenvalue of the empirical NTK.

**Assumption 3.3** (Positive definiteness of NTK). There exists $c_{\text{pos}} > 0$ such that $\mu_{\min}(K_m^{(t)}) \geq c_{\text{pos}}$ for all $t \in [0, T]$, and there exists $c_\infty > 0$ such that $\mu_{\min}(K_\infty) \geq c_\infty$. The constants $c_{\text{pos}}, c_\infty$ are independent of $m, n$ in the regime of interest.

Assumption 3.3 is supported by the positive definiteness of the NTK in the wide networks (Nguyen et al., 2021).

**Assumption 3.4** (Upper bound on diagonal kernel under input normalization). Assume inputs satisfy $\|\boldsymbol{x}\|_2 \leq 1$ for all $\boldsymbol{x}$ considered. Then there exist finite constants $K_\infty^{\max}, K_m^{\max} > 0$ such that

$$K_\infty(\boldsymbol{x}, \boldsymbol{x}) \leq K_\infty^{\max}, \qquad \sup_{t \in [0, T]} K_m^{(t)}(\boldsymbol{x}, \boldsymbol{x}) \leq K_m^{\max}.$$

Assumption 3.4 is a benign condition, typically satisfied in practice with normalized inputs.

**Assumption 3.5** ($\mu_{\min}$ ratio). There exist universal constants $c_-, c_+, c_\infty > 0$ such that

$$c_- \mu_{\min}(K_m^{(0)}) \leq \mu_{\min}(K_m^{(T)}) \leq c_+ \mu_{\min}(K_m^{(0)}), \qquad \mu_{\min}(K_\infty) \geq c_\infty \mu_{\min}(K_m^{(0)}).$$

Assumption 3.5 implies that the smallest eigenvalue remains up to a constant factor with respect to time and width. This is empirically verified in Appendix B.

**Assumption 3.6** (Kernel drift)**.** There exist positive constants $C_{\mathrm{drift}}$ and $C_{\mathrm{init}}$, and a non-increasing function $\phi(m) \downarrow 0$ such that,

$$\sup_{u \in [0,T]} \big\| K_m^{(u)} - K_m^{(0)} \big\| \ \leq \ C_{\mathrm{drift}}\, \phi(m), \qquad \big\| K_m^{(0)} - K_\infty \big\| \ \leq \ C_{\mathrm{init}}\, \phi(m).$$

Under the standard NTK conditions of Lee et al. (2019), one may take $\phi(m) = m^{-1/2}$ for any depth of deep neural networks, but the rate is not required here. Note that we discuss the operator norm in this assumption for proofs later, instead of the Frobenius norm, which is used in Lee et al. (2019). However, the operator norm is smaller than the Frobenius norm, so directly replacing our results on the operator norm to the Frobnius norm is valid. Assumption 3.6 captures the kernel stability in lazy training and is empirically validated in Appendix B.

Then, we derive the key theorem in this paper: the generalization error of a finite-width network $E_g^{(m)}$ by evaluating $|E_g^{(m)} - E_g^{(\infty)}|$ via the smallest eigenvalue of the empirical NTK.

**Theorem 3.7** (Finite-width test error via the empirical NTK)**.** *Let Assumptions 3.3–3.5 hold. Then,*

$$E_g^{(m)} \ \leq \ E_g^{(\infty)} \ + \ C_3\, \frac{\left( \sup_u \| K_m^{(u)} - K_m^{(0)} \| + \| K_m^{(T)} - K_m^{(0)} \| + \| K_m^{(0)} - K_\infty \| \right)}{\mu_{\min}\big(K_m^{(0)}\big)^2}, \qquad (8)$$

*where the constant $C_3$ is independent of $m$ or $n$.*

This theorem evaluates the difference between the generalization error of finite-width and infinite-width networks via the smallest eigenvalue and the norm of the empirical NTK. Note that it does not assume lazy training like Assumption 3.6. Intuitively, the functional difference $|E_g^{(m)} - E_g^{(\infty)}|$ is small in lazy training since the NTK is stable during the training, whereas it becomes larger in feature learning since the kernel change $\| K_m^{(T)} - K_m^{(0)} \|$ becomes large in the regime.

**Theorem 3.8** (Finite-width test error in lazy training)**.** *Under Assumptions 3.1, 3.3–3.6, the test error $E_g^{(m)}$ with constants $C_4$ and $C_5$ independent of $m$ satisfies*

$$E_g^{(m)} \ \leq \ \frac{C_4}{\mu_{\min}\big(K_m^{(0)}\big)^2} \ + \ C_5\, \frac{\phi(m)}{\mu_{\min}\big(K_m^{(0)}\big)^2}. \qquad (9)$$

This is a special case of Theorem 3.7, which applies Assumption 3.6 representing the kernel change during training with the width $m$ in sufficiently wide networks. In Theorem 3.7, an upper bound on $E_g^{(m)}$ is governed by the smallest eigenvalue of the empirical NTK at initialization, with a width-dependent correction $\phi(m)$ that vanishes as $m \to \infty$. This theorem establishes that a single quantity, $\mu_{\min}$ of the empirical NTK at initialization, controls not only the infinite-width but also the finite-width generalization. These results also indicate that the generalization error decreases as the sufficiently large width $m$ increases. It is observed in our experiment that $\mu_{\min}$ increases as the width increases, and the increase saturates at a sufficiently large width. This shows that $E_g^{(m)}$ decreases in wider networks, but the decrease saturates at sufficiently large networks. Practically, this means we can decide when widening a network no longer yields effective improvement by monitoring the saturation of $\mu_{\min}\big(K_m^{(0)}\big)$ across widths.

**Note on theoretical novelty.** Although the smallest eigenvalue of the NTK is already recognized as important for generalization, to the best of our knowledge, this is the first work to explicitly characterize the direct relationship between generalization error and the smallest eigenvalue of the finite-width NTK. For example, combining optimization analysis under PL condition (Liu et al., 2022) and PL-based generalization bound (Lei & Ying, 2021) in the NTK regime implies the relationship between generalization error and the smallest eigenvalue of the NTK. However, the combination requires resolving nontrivial mismatches between their assumptions, highlighting that our contribution is to provide an explicit and self-contained description of this relationship.

## 3.3 SKETCH OF PROOF

In this section, we provide a sketch of proof for Theorems 3.2, 3.7, and 3.8. A detailed proof can be found in Appendix A. We start the argument by introducing the necessary tools.

**Tool 1: Mercer expansions.** On the empirical measure $p_n = \frac{1}{n} \sum_{j=1}^{n} \delta_{\boldsymbol{x}_j}$, let $G := K(X, X) \in \mathbb{R}^{n \times n}$ with eigenpairs $G u_k = \mu_k u_k$ and $u_k^\top u_{k'} = \delta_{kk'}$, and let us define $\phi_k(\boldsymbol{x}_j) := \sqrt{n}\, u_k[j]$. Then, on $\mathrm{supp}(p_n)$,

$$K(\boldsymbol{x}, \boldsymbol{x}') = \sum_{k=1}^{n} \frac{\mu_k}{n}\, \phi_k(\boldsymbol{x}) \phi_k(\boldsymbol{x}'). \tag{10}$$

Equation 10 is the finite-sample Mercer expansion obtained by replacing the population measure $p_X$ in the classical Mercer theorem Minh et al. (2006) with $p_n$ and working with the Gram spectrum of $G$. In the population statement, the kernel expands in eigenfunctions $\{\phi_\rho\}_{\rho \geq 1}$ of the integral operator $(\mathcal{T}_K f)(\boldsymbol{x}) = \int p_X(\boldsymbol{x}') K(\boldsymbol{x}, \boldsymbol{x}') f(\boldsymbol{x}')\, d\boldsymbol{x}'$ with eigenvalues $\{\lambda_\rho\}_{\rho \geq 1}$. The discrete version above uses the $L^2(p_n)$ basis induced by $\{u_k\}$ and empirical eigenvalues $\{\mu_k\}_{k=1}^{n}$. We adopt Equation 10 to rewrite the following test error predicted by Canatar et al. (2021) into a finite-dataset form.

**Tool 2: Closed-form test error for KRR.** Specializing Canatar et al. (2021) to a finite dataset and the empirical Mercer basis in the Tool 1, the test error $E_g$ of KRR is expressed as:

$$E_g = \frac{1}{1 - \gamma} \sum_{k=1}^{n} \frac{\mu_k}{n\, (\kappa + \mu_k)^2} \left( \kappa^2 w_k^2 + \sigma^2 \mu_k \right), \tag{11}$$

with fixed-point scalars

$$\kappa = \alpha + \frac{\kappa}{n} \sum_{k=1}^{n} \frac{\mu_k}{\kappa + \mu_k}, \qquad \gamma = \frac{1}{n} \sum_{k=1}^{n} \frac{\mu_k^2}{(\kappa + \mu_k)^2}. \tag{12}$$

All symbols follow prior definitions. Here $\mu_k$ are the Gram eigenvalues from Equation 10, $n$ is the sample size, $\alpha$ is the ridge, and $\sigma^2$ is the noise variance. The new quantities are $\kappa$, the effective ridge determined by the fixed point in Equation 12, $\gamma \in [0, 1)$, a shrinkage factor that increases toward the interpolation peak, and $w_k$, the target weights in the empirical basis associated with Equation 10. Equation 11 is regarded as a mode-wise bias–variance decomposition in the Gram spectrum. $\mu_k/(\kappa + \mu_k)^2$ acts as a spectral filter, $\kappa^2 w_k^2$ contributes bias, and $\sigma^2 \mu_k$ contributes variance. The prefactor $(1 - \gamma)^{-1}$ amplifies risk as $\gamma \to 1$, quantifying proximity to interpolation.

**Theorem 3.2 (infinite width).** Write the generalization error $E_g$ of a kernel regression model by Canatar et al. (2021):

$$E_g = \frac{1}{1 - \gamma}(B + V)$$

$$B = \sum_{k=1}^{n} \frac{\mu_k}{n} \frac{\kappa^2 w_k^2}{(\kappa + \mu_k)^2}, \qquad V = \sum_{k=1}^{n} \frac{\sigma^2 \mu_k^2}{n(\kappa + \mu_k)^2}.$$

As for the first term, since the function $x \mapsto \left(\frac{\kappa}{\kappa + x}\right)^2$ monotonically decreases on $[0, \infty)$,

$$B \leq \frac{\kappa^2}{n(\kappa + \mu_{\min})^2} \sum_{k=1}^{n} \mu_k w_k^2 = \frac{\kappa^2}{(\kappa + \mu_{\min})^2} \|f^*\|_{L^2(p_n)}^2.$$

In the ridgeless case, $\kappa \leq \frac{1}{n} \mathrm{tr}(G) \leq K_\infty := \sup_{\boldsymbol{x}} K(\boldsymbol{x}, \boldsymbol{x})$ and, away from the interpolation peak (Assumption 3.1), $1 - \gamma = \Theta(1)$. Hence, the inequality below holds using constant $C_1$:

$$\frac{B}{1 - \gamma} \leq C_1\, \mu_{\min}^{-2}.$$

Regarding the second term, from Equation 12,

$$\gamma = \frac{1}{n} \sum_k \frac{\mu_k^2}{(\kappa + \mu_k)^2} \leq \frac{1}{(\kappa + \mu_{\min})^2} \cdot \frac{1}{n} \sum_k \mu_k^2 \leq \frac{n K_\infty^2}{(\kappa + \mu_{\min})^2}.$$

Therefore, using constant $C_2$

$$\frac{V}{1 - \gamma} \ \leq \ C_2 \, \frac{\sigma^2 \, n}{\mu_{\min}^2}.$$

Combining $B$ and $V$ gives Theorem 3.2.

**Theorem 3.7 (finite width).** Let $f_m(\cdot, t)$ be the width-$m$ network, $K_m^{(t)}$ be its empirical NTK on $X$, and $\widehat{f}_m^{(t)}$ be the ridgeless regressor for $K_m^{(t)}$. With $f_\infty$ the regressor for $K_\infty$,

$$|f_m(T) - f_\infty| \ \leq \ \underbrace{|f_m(T) - \widehat{f}_m^{(T)}|}_{\text{(G1)}} + \underbrace{|\widehat{f}_m^{(T)} - \widehat{f}_m^{(0)}|}_{\text{(G2)}} + \underbrace{|\widehat{f}_m^{(0)} - f_\infty|}_{\text{(G3)}}. \tag{13}$$

Using Duhamel's principle for (G1) and the closed solution of KRR for (G2)–(G3), under Assumptions 3.3–3.5 we obtain

$$\begin{cases} |f_m(T) - \widehat{f}_m^{(T)}| \leq C \, \dfrac{\sup_{u \in [0,T]} \|K_m^{(u)} - K_m^{(T)}\|}{\mu_{\min}(K_m^{(0)})} & \text{(G1)} \\[2ex] |\widehat{f}_m^{(T)} - \widehat{f}_m^{(0)}| \leq \dfrac{C_{\text{tv}}}{\mu_{\min}(K_m^{(0)})^2} \, \|K_m^{(T)} - K_m^{(0)}\| & \text{(G2)} \\[2ex] |\widehat{f}_m^{(0)} - f_\infty| \leq \dfrac{C_{\text{init}}}{\mu_{\min}(K_m^{(0)})^2} \, \|K_m^{(0)} - K_\infty\| & \text{(G3)} \end{cases}$$

These inequalities are described in formal proofs. Combining (G1)–(G3) gives

$$|f_m(T) - f_\infty| \ \leq \ \frac{C_*}{\mu_{\min}(K_m^{(0)})^2} \Big( \sup_u \|K_m^{(u)} - K_m^{(0)}\| + \|K_m^{(T)} - K_m^{(0)}\| + \|K_m^{(0)} - K_\infty\| \Big). \tag{14}$$

Define $E_g^{(m)} = \mathbb{E}_{(X, \boldsymbol{y}) \sim \mathcal{D}}[(f(\boldsymbol{x}) - f_m(\boldsymbol{x}, T))^2]$ and $E_g^{(\infty)}$ analogously. By Cauchy–Schwarz inequalities,

$$E_g^{(m)} - E_g^{(\infty)} \leq C_3 \, |f_m(T) - f_\infty|. \tag{15}$$

Assumption 3.5 lets us express the order in $\mu_{\min}^{-2}(K_m^{(0)})$, yielding Theorem 3.7.

**Theorem 3.8 (Finite-width test error in lazy training).** Under lazy-training stability of Assumption 3.6, each kernel difference is $\phi(m)$, so the second term of Equation 14 is smaller than $C \, \phi(m) / \mu_{\min}(K_m^{(0)})^2$. The first term comes from Theorem 3.2.

### 3.4 EXPERIMENTAL VERIFICATION

We now present a series of experimental results to verify our key theorem, Theorem 3.8 in a two-layer network. In Figure 1, we train two-layer fully connected networks with a ReLU activation function on the following synthetic data generated via `make_regression`[1] from `scikit-learn` library while changing the width of the network in the range: $\{16, 32, \dots, 1024\}$. The dataset comprises 2,000 samples, each with 20 input features, and includes Gaussian noise with a standard deviation of 10.0. All the weights are initialized as what Lee et al. call the NTK parameterization. Then, the models are trained with optimizer SGD, learning rate 0.001, and batch size 32 for 1000 epochs.

The top plots in Figure 1 establish that both test loss and the NTK's smallest eigenvalue stabilize at sufficiently large width. We therefore treat the largest width in our sweep as the asymptotic anchor. In the bottom plot, from that point (star), a slope-1 line corresponds to the theoretical bound $E_g = \mathcal{O}(\mu_{\min}^{-2})$. All empirical points lie below this black dashed line across widths, consistent with Theorem 3.8. At the same time, the trend of the scatter is noticeably shallower and is well captured by a red dashed line. In short, the data respect the worst-case bound while exhibiting a milder, approximately $1/\sqrt{\mu_{\min}}$ dependence in practice.

---

[1]The document is at: `https://scikit-learn.org/stable/modules/generated/sklearn.datasets.make_regression.html`

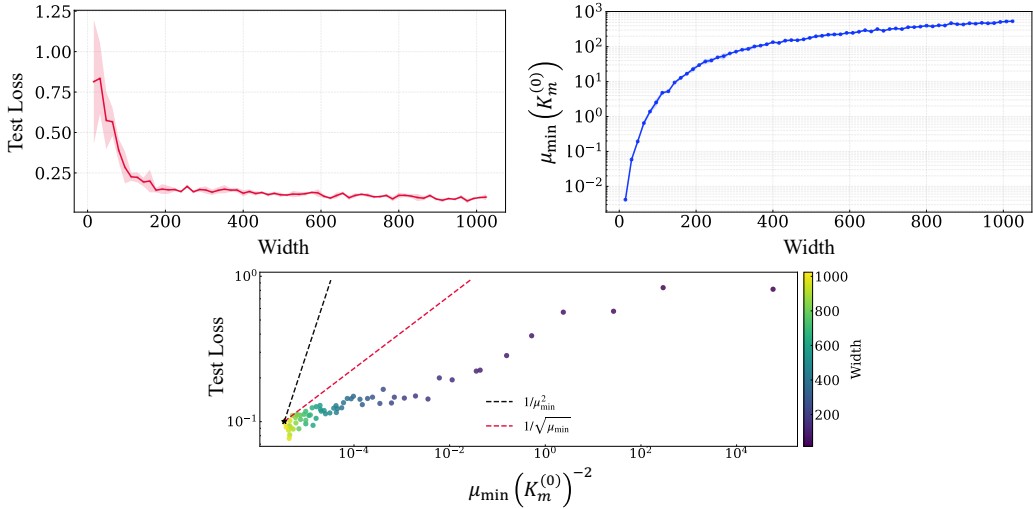

Figure 1: $\mu_{\min}^{-2}$ **and test loss on two-layer networks.** From the top plots, we see that the decrease in the test loss and the increase in $\mu_{\min}$ saturate in wide networks. The bottom plot illustrates that the test loss is upper bounded by $\mathcal{O}(\mu_{\min}^{-2})$ (consistent with Theorem 3.8) but is closer to $\mathcal{O}(1/\sqrt{\mu_{\min}})$. The scope of this plot is the point with sufficiently large width (closer to yellow).

## 4    METHOD FOR THE CARDINAL WIDTH

In this section, we propose a method to determine the *cardinal width*. The theoretical and empirical results in Section 3 show that test error is governed by the smallest NTK eigenvalue. Leveraging this, we read off the width at which the minimum eigenvalue saturates and take it as a training-free proxy for the width where the test error saturates, i.e., the *cardinal width*.

### 4.1    ALGORITHM

We now present our algorithm for determining the cardinal width. See Algorithm 1 for pseudocode. First, given a fixed architecture form with flexible width, dataset $X = \{x_i\}_{i=1}^n$, and initialization scheme $\mathcal{I}$, we scan candidate widths $m \in \mathcal{M}$. At each width, we compute the empirical NTK at initialization to estimate its smallest eigenvalue. For this step, we use Locally Optimal Block Pre-conditioned Conjugate Gradient (LOBPCG) (Knyazev, 2001), which finds a few extreme eigenpairs by minimizing the Rayleigh quotient over small subspaces. The resulting scatter $\{(m, \widehat{\mu}_{\min}(m))\}$ exhibits a saturating trend as $m$ grows (see Figure 3). To make this trend operational, we fit a simple saturating curve:

$$g(x) = -\frac{ax}{b+x} + c \tag{16}$$

by least squares. We emphasize, however, that this selection does not exclude the possibility of alternative functional forms which could also provide a suitable fit. We then declare the cardinal width as the smallest $m$ at which the fitted slope becomes negligible.

### 4.2    EXPERIMENTS

We evaluate the proposed method by applying it across several architecture–dataset pairs. In concrete, we train deep neural networks (DNNs) on the UCI Diabetes and California Housing datasets, and CNN and ResNet (He et al., 2016) on MNIST (Lecun et al., 1998) and CIFAR-10 (Krizhevsky & Hinton, 2009). The term *width* refers to the number of channels in convolutional layers (Arora et al., 2019) in CNNs and the number of neurons at each layer in ResNet (Huang et al., 2020). For each pair, we follow Algorithm 1 to compute the cardinal width. We then train the same models across a range of widths under a fixed training protocol and draw a test loss curve with respect to width.

---

**Algorithm 1** Training-free cardinal width via the smallest eigenvalue of the NTK

---

**Inputs:** Architecture (fixed form), dataset $X = \{x_i\}_{i=1}^n$ (labels not used), initialization $\mathcal{I}$, width grid $\mathcal{M} \subset \mathbb{N}$, threshold $\delta > 0$, fitting family $g(m; \vartheta) = -\frac{am}{b+m} + c$

**Output:** Cardinal width $m_{\text{card}}$

1: **for** $m \in \mathcal{M}$ **do**
2: $\quad \theta_0 \leftarrow \mathcal{I}(m)$
3: $\quad$ Compute empirical NTK at initialization: $(K_m^{(0)})_{ij} = \nabla_\theta f_{\theta_0}(x_i)^\top \nabla_\theta f_{\theta_0}(x_j)$
4: $\quad$ Estimate the smallest eigenvalue of $K_m^{(0)}$ using LOBPCG and record $\widehat{\mu}_{\min}(m)$
5: **end for**
6: Fit $g$ to $\{(m, \widehat{\mu}_{\min}(m))\}$: $\hat{\vartheta} \leftarrow \arg\min_\vartheta \sum_{m \in \mathcal{M}} (\widehat{\mu}_{\min}(m) - g(m; \vartheta))^2$
7: Define $\hat{g}(m) := g(m; \hat{\vartheta})$ and compute its derivative $d\hat{g}(m)/dm$
8: $m_{\text{card}} \leftarrow \min\{ m \in \mathcal{M} : \frac{d}{dm}\hat{g}(m) < \delta \}$
9: **return** $m_{\text{card}}$

---

In this paper, we focus on regression tasks, so these models predict the exact value of the index. Evaluation proceeds by comparing the predicted cardinal width with the empirical saturation point where additional widening yields negligible improvement. For all the experiments, both $\mu_{\min}$ and the test loss are computed over five independent runs, and the reported results are then averaged. We provide experimental details such as architectures and hyperparameters for training in Appendix C.

Figure 2 illustrates that the behavior of $\mu_{\min}$ as a function of network width $m$ closely aligns with the trend observed in the test loss across all architectures. The green vertical lines indicate the widths at which the derivative of the fitted function $g$ drops below the predefined threshold $\delta$. These points correspond to the saturation of the smallest eigenvalue, and notably, they also coincide with the point where the test loss ceases to improve significantly. This alignment supports the validity of $\mu_{\min}$ as a proxy for identifying the cardinal width prior to training. Below are some notes about the results.

**Note 1: Lazy training / feature learning.** The experiments are conducted without any constraints on lazy training (e.g., step sizes or initialization methods). Theorem 3.8 formally applies to the lazy training regime and therefore the theoretical validity of $\mu_{\min}$ as a proxy in determining width does not hold in feature learning. Interestingly, however, the saturation of $\mu_{\min}$ coincides with the saturation of test loss in the model operating in the feature learning regime. This result indicates that the smallest eigenvalue of the NTK can act as a robust indicator of the cardinal width even when features evolve during training.

We assume lazy training to derive Theorem 3.8 for theoretical rigor, but potential explanation that the theorem holds outside lazy training can be provided. Bordelon & Pehlevan (2023) show that in the Dynamical Mean Field Theory (DMFT), the fluctuations over random initialization of NTK are of order $\mathcal{O}(m^{-1/2})$, and the variance is $\mathcal{O}(1/m)$ for a finite width $m$ even in the feature-learning regime. The authors show that richer feature learning leads to closer agreement with infinite-width mean field behavior. Thus, the kernel change $\|K_m^{(T)} - K_m^{(0)}\|$ would converge to a constant (not necessarily zero as in lazy training). We assume lazy training to ensure $(\sup_u \|K_m^{(u)} - K_m^{(0)}\| + \|K_m^{(T)} - K_m^{(0)}\| + \|K_m^{(0)} - K_\infty\|) \to 0$ in sufficiently wide networks and derive Theorem 3.8. However, even in the feature-learning regime, this part becomes a small constant in sufficiently wide networks if we apply the result of Bordelon & Pehlevan (2023). Therefore, the main message of our theory that "the smallest eigenvalue of the NTK controls the upper bound of generalization error" remains valid in feature learning. Thus, $\mu_{\min}$ would work as an indicator for cardinal width in both lazy-training and feature-learning regimes.

**Note 2: Fluctuation among training recipes.** Since the NTK depends on the architecture, dataset, and initialization, our width-selection problem is posed after these choices are fixed. In other words, the recommended width is determined per-architecture, per-dataset, and per-initialization. While the appropriate width would differ among training recipes (optimizer, learning rate, etc.), empirically, the cardinal width varies little across such training recipes (See Appendix D.1). Thus, our predicted saturation width remains broadly stable over practical parameters.

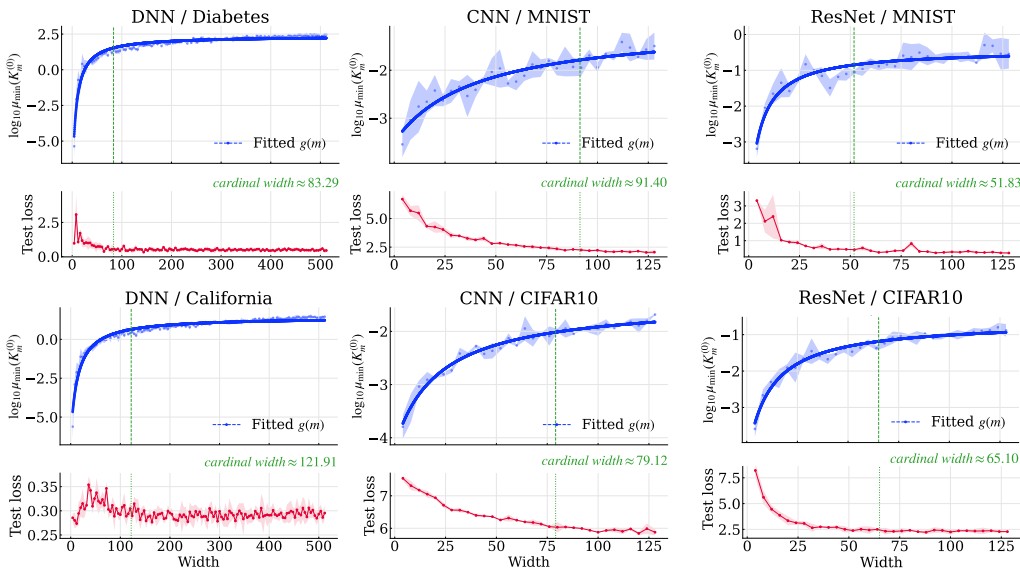

Figure 2: **The cardinal width identified using fitted $\mu_{\min}$.** Green lines indicate where the growth of $\mu_{\min}$ slows down, suggesting a saturation point. At these widths, the test loss also plateaus, validating the proposed criterion for determining the cardinal width.

**Note 3: Computational cost.** While the NTK computations can be expensive, our procedure only requires the *smallest* eigenvalue of the initialization NTK per width, which we obtain with the iterative eigensolver LOBPCG. For example, on a DNN with California Housing (with about 20K samples), one estimate of $\mu_{\min}$ per width finishes in on the order of a minute, well within a realistic cost. In addition, computing $\mu_{\min}$ on a subset further reduces cost; we observe that the *relative* saturation across widths is stable under moderate subsampling. See Appendix D.2.

## 5 CONCLUSION AND LIMITATIONS

We propose a training-free method for estimating the cardinal width of a neural network by tracking the smallest eigenvalue $\mu_{\min}$ of the NTK at initialization. We theoretically show that in the infinite-width regime, the test error admits an upper bound proportional to $\mu_{\min}^{-2}$, and for sufficiently wide finite models, we obtain an analogous proportionality via the empirical NTK, elevating the spectral edge to a practical predictor of generalization. By leveraging this relationship, we design a practical metric that identifies the width where further increases no longer yield significant generalization benefits. We validate our approach on multiple datasets and architectures, illustrating that the saturation point of the smallest eigenvalue of the NTK corresponds well with the point of test loss saturation. Our method enables model designers to determine the cardinal width with theoretical foundations and without any training, thus saving intensive trial-and-error.

Our main limitations can however be seen in the theoretical constraints, i.e., the theoretical results are derived under the lazy training regime with squared loss and gradient flow, and thus may not directly extend to settings involving strong feature learning, alternative losses, or practical optimizers. In addition, our results are largely restricted to regression tasks, leaving open questions about the applicability of the approach to classification tasks. While the NTK can also be computed for classification problems and extending our method to this setting is technically feasible, it lacks a theoretical guarantee at present. A natural next step is to extend the analysis to theory on feature learning and classification problems.

## REPRODUCIBILITY STATEMENT

The experimental settings are fully described in Section 3, Section 4, and Appendix C. Our code is available at `https://github.com/Suna-D/cardinal-width`.

## ACKNOWLEDGEMENTS

The authors thank Matteo Gamba for helpful discussions during the early development of this work. We also thank the anonymous reviewers for their constructive feedback.

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

APPENDIX

In this appendix, we provide the proof of Theorem 3.2, 3.7, and 3.8 in Appendix A, along with the experimental verification of the assumptions in the proof. Details on the experiment for estimating the cardinal width are provided in Appendix C. Then, supplementary notations are given in Appendix D. Further, we will discuss the extension of the proposed method to classification tasks in Appendix E. Finally, we provide our LLM usage in Appendix F.

## A    FORMAL PROOFS

In this section, we provide formal proofs of the theorems in the main paper. We develop the proofs in a single flow that mirrors the setup in Sec. 3.1. Beginning from kernel ridge (less) regression, we first make Mercer expansions operational on a finite dataset. We then recall the closed-form generalization error of Canatar et al. (2021), specialize it to the finite-data Mercer basis, and use it to prove Theorem 3.2. Finally, we pass from the infinite-width predictor to finite-width networks trained by gradient flow, proving Theorems 3.7 and 3.8. Throughout, all symbols in this section follow the definitions in the main paper.

### A.1    DETAILS OF MAIN TOOLS

**Tool 1: Mercer's theorem.**    On the population distribution $p_X$, define the integral operator

$$(\mathcal{T}_K f)(\boldsymbol{x}) = \int p_X(\boldsymbol{x}')\, K(\boldsymbol{x}, \boldsymbol{x}')\, f(\boldsymbol{x}')\, d\boldsymbol{x}'.$$

Assume $\mathcal{T}_K$ has $L^2(p_X)$–orthonormal eigenfunctions $\{\varphi_\rho\}_{\rho \geq 1}$ with eigenvalues $\{\lambda_\rho\}_{\rho \geq 1}$. Mercer's theorem gives

$$K(\boldsymbol{x}, \boldsymbol{x}') = \sum_{\rho \geq 1} \lambda_\rho\, \varphi_\rho(\boldsymbol{x})\, \varphi_\rho(\boldsymbol{x}') \quad \text{in } L^2(p_X). \tag{17}$$

We work with a finite dataset $X = \{\boldsymbol{x}_j\}_{j=1}^n$ and replace $p_X$ by the empirical measure $p_n = \frac{1}{n} \sum_{j=1}^n \delta_{\boldsymbol{x}_j}$. Let $G := K(X, X) \in \mathbb{R}^{n \times n}$ and write its eigenpairs as $G u_k = \mu_k u_k$ with $u_k^\top u_{k'} = \delta_{kk'}$. Induce an $L^2(p_n)$–orthonormal basis on $\mathrm{supp}(p_n)$ by

$$\phi_k(\boldsymbol{x}_j) = \sqrt{n}\, u_k[j], \qquad \frac{1}{n} \sum_{j=1}^n \phi_k(\boldsymbol{x}_j)\phi_{k'}(\boldsymbol{x}_j) = \delta_{kk'}.$$

Define $\lambda_k$ as $\lambda_k := \mu_k/n$. The kernel then admits the expansion to the finite number of data:

$$K(\boldsymbol{x}, \boldsymbol{x}') = \sum_{k=1}^n \lambda_k\, \phi_k(\boldsymbol{x})\, \phi_k(\boldsymbol{x}') \quad \text{on } \mathrm{supp}(p_n), \tag{18}$$

which is the empirical counterpart of Equation 17 under $p_n$.

Define the feature functions

$$\psi_k(\boldsymbol{x}) = \sqrt{\lambda_k}\, \phi_k(\boldsymbol{x}) \quad \text{so that} \quad \psi_k(\boldsymbol{x}_j) = \sqrt{\mu_k}\, u_k[j].$$

We expand the target and the learned predictor as

$$f^*(\boldsymbol{x}) = \sum_{k=1}^n w_k^*\, \psi_k(\boldsymbol{x}), \qquad \hat{f}(\boldsymbol{x}) = \sum_{k=1}^n \overline{w}_k\, \psi_k(\boldsymbol{x}).$$

**Tool 2: The generalization error of Canatar et al. (2021)**    Specializing Canatar et al. (2021) to our notation, the generalization error of KRR with ridge $\alpha \geq 0$ can be written in the Mercer basis as

$$E_g = \frac{1}{1 - \gamma} \sum_i \frac{\lambda_i}{(\kappa + n\lambda_i)^2} \left(\kappa^2 w_i^2 + \sigma^2 n\lambda_i\right),$$

$$\kappa = \alpha + \sum_i \frac{\kappa\lambda_i}{\kappa + n\lambda_i}, \qquad \gamma = \sum_i \frac{n\lambda_i^2}{(\kappa + n\lambda_i)^2}. \tag{19}$$

Intuitively, $\lambda_k/(\kappa + n\lambda_k)^2$ act as spectral filters. The directions with $n\lambda_k \gg \kappa$ are shrunk, while directions with small $\lambda_k$ are relatively amplified. Using the finite-data expansion Equation 18 and $\lambda_k = \frac{\mu_k}{n}$, Equation 19 can be expressed using empirical eigenvalues in the main paper:

$$E_g = \frac{1}{1-\gamma}\sum_{k=1}^{n}\frac{\mu_k}{n\,(\kappa+\mu_k)^2}\Big(\kappa^2 w_k^2 + \sigma^2\,\mu_k\Big),$$

$$\kappa = \alpha \;+\; \frac{\kappa}{n}\sum_{k=1}^{n}\frac{\mu_k}{\kappa+\mu_k},\; \gamma = \frac{1}{n}\sum_{k=1}^{n}\frac{\mu_k^2}{(\kappa+\mu_k)^2}. \tag{20}$$

## A.2 PROOF OF THEOREM 3.2 (INFINITE WIDTH)

Under Assumption 3.1, the KRR model is away from the interpolation peak, so there exists $c_0 \in (0,1)$ with $1-\gamma \ge c_0$. Decompose $E_g$ of Equation 20 into bias $B$ and variance $V$ terms,

$$E_g \;=\; \frac{1}{1-\gamma}\,(B+V), \qquad B = \sum_{k=1}^{n}\frac{\mu_k}{n}\frac{\kappa^2 w_k^2}{(\kappa+\mu_k)^2}, \quad V = \sum_{k=1}^{n}\frac{\sigma^2\mu_k^2}{n(\kappa+\mu_k)^2}.$$

**Bias.** The function $x \mapsto (\frac{\kappa}{\kappa+x})^2$ is monotonically decreasing for $x \ge 0$, hence

$$B \;\le\; \frac{\kappa^2}{n(\kappa+\mu_{\min})^2}\sum_{k=1}^{n}\mu_k w_k^2 \;=\; \frac{\kappa^2}{(\kappa+\mu_{\min})^2}\,\|f^*\|_{L^2(p_n)}^2.$$

In the ridgeless case $\alpha = 0$, the fixed point gives $\kappa \le \frac{1}{n}\mathrm{tr}(G) \le K_\infty$. With $1-\gamma \ge c_0$, we obtain

$$\frac{B}{1-\gamma} \;\le\; \frac{1}{c_0}\cdot\frac{\kappa^2}{(\kappa+\mu_{\min})^2}\,\|f^*\|_{L^2(p_n)}^2 \;\le\; C_1\,\mu_{\min}^{-2}. \tag{21}$$

**Variance.** From Equation 20,

$$\gamma \;=\; \frac{1}{n}\sum_{k=1}^{n}\frac{\mu_k^2}{(\kappa+\mu_k)^2} \;\le\; \frac{1}{(\kappa+\mu_{\min})^2}\cdot\frac{1}{n}\sum_{k=1}^{n}\mu_k^2 \;=\; \frac{\mathrm{tr}(G^2)}{n(\kappa+\mu_{\min})^2}.$$

By the Cauchy–Schwarz bound $K(\boldsymbol{x}_i,\boldsymbol{x}_j)^2 \le K(\boldsymbol{x}_i,\boldsymbol{x}_i)K(\boldsymbol{x}_j,\boldsymbol{x}_j) \le K_\infty^2$, we have $\mathrm{tr}(G^2) = \sum_{i,j}K(\boldsymbol{x}_i,\boldsymbol{x}_j)^2 \le n^2 K_\infty^2$ and thus $\mathrm{tr}(G^2)/n \le nK_\infty^2$. Therefore,

$$\frac{V}{1-\gamma} \;=\; \sigma^2\frac{\gamma}{1-\gamma} \;\le\; \frac{\sigma^2}{c_0}\cdot\frac{\mathrm{tr}(G^2)}{n(\kappa+\mu_{\min})^2} \;\le\; C_2\,\frac{\sigma^2 n}{\mu_{\min}^2}. \tag{22}$$

Combining Equation 21 and Equation 22 yields Equation 6 and proves Theorem 3.2.

## A.3 PROOF OF THEOREM 3.7 (FROM INFINITE TO FINITE WIDTH)

We now relate the finite-width network trained by gradient flow to infinite-width network (kernel ridgeless regressor). Let $f_m(\cdot, t)$ be the width-$m$ network at time $t \in [0, T]$, and $K_m^{(t)}$ its empirical NTK Gram on $X$. Write $\widehat{f}_m^{(t)}$ for the *ridgeless* KRR predictor built from $K_m^{(t)}$, and let $f_\infty$ be the NTK regressor associated with $K_\infty$. Our goal here is to evaluate $|f_m(T) - f_\infty|$.

**A three-term decomposition.** Using the triangle inequality, we decompose $|f_m(T) - f_\infty|$ into three terms and derive an upper bound for each term:

$$|f_m(T) - f_\infty| \;\le\; \underbrace{|f_m(T) - \widehat{f}_m^{(T)}|}_{\text{(G1) linearization gap}} + \underbrace{|\widehat{f}_m^{(T)} - \widehat{f}_m^{(0)}|}_{\text{(G2) time-variation gap}} + \underbrace{|\widehat{f}_m^{(0)} - f_\infty|}_{\text{(G3) finite vs. infinite width}}. \tag{23}$$

We now bound (G1)–(G3) under Assumptions 3.3–3.5.

**(G1) Linearization gap.** Let $\boldsymbol{F}(t) \in \mathbb{R}^n$ be the vector of training predictions at time $t$, and define the difference $\boldsymbol{r}(t) := \boldsymbol{F}(t) - \boldsymbol{y}$. The gradient flow dynamics on the training points satisfy

$$\frac{d}{dt}\boldsymbol{r}(t) = -K_m^{(t)}\boldsymbol{r}(t), \qquad \boldsymbol{r}(0) = \boldsymbol{F}(0) - \boldsymbol{y}. \tag{24}$$

Fix $T > 0$ and introduce the residual $\boldsymbol{r}_T(t)$ of the frozen kernel $K_m^{(T)}$ as the unique solution of

$$\frac{d}{dt}\boldsymbol{r}_T(t) = -K_m^{(T)}\boldsymbol{r}_T(t), \qquad \boldsymbol{r}_T(0) = \boldsymbol{r}(0), \tag{25}$$

so that the solution of Equation 25 is:

$$\boldsymbol{r}_T(t) = e^{-K_m^{(T)}t}\boldsymbol{r}(0). \tag{26}$$

Define the deviation $\boldsymbol{h}(t) := \boldsymbol{r}(t) - \boldsymbol{r}_T(t)$. Subtracting Equation 24 and 25 above yields

$$\frac{d}{dt}\boldsymbol{h}(t) = -K_m^{(T)}\boldsymbol{h}(t) - \big(K_m^{(t)} - K_m^{(T)}\big)\boldsymbol{r}(t), \qquad \boldsymbol{h}(0) = \boldsymbol{0}. \tag{27}$$

By applying Duhamel's principle to Equation 27, we obtain

$$\boldsymbol{h}(t) = \int_0^t e^{-K_m^{(T)}(t-s)}\big(K_m^{(T)} - K_m^{(s)}\big)\boldsymbol{r}(s)\,ds. \tag{28}$$

Since $K_m^{(T)}$ is positive semidefinite, $\|e^{-K_m^{(T)}(t-s)}\| \le 1$ for $t \ge s$. Thus,

$$\begin{aligned}
\|\boldsymbol{h}(t)\| &\le \int_0^t \big\|K_m^{(T)} - K_m^{(s)}\big\| \, \|\boldsymbol{r}(s)\| \, ds \\
&\le \Big(\sup_{u\in[0,T]} \|K_m^{(u)} - K_m^{(T)}\|\Big) \int_0^t |\boldsymbol{r}(s)|\,ds.
\end{aligned} \tag{29}$$

For a test input $\boldsymbol{x}$, let $\boldsymbol{k}^{(t)}(\boldsymbol{x}) \in \mathbb{R}^n$ be the kernel vector with entries $K_m^{(t)}(\boldsymbol{x}, \boldsymbol{x}_i)$ and $\tilde{f}_T(\boldsymbol{x}, \cdot)$ be a ridgeless KRR model with the frozen $k^{(T)}(\boldsymbol{x})$. The function space dynamics obey

$$\begin{aligned}
\frac{d}{dt}f_m(\boldsymbol{x}, t) &= -\boldsymbol{k}^{(t)}(\boldsymbol{x})^\top \boldsymbol{r}(t), \\
\frac{d}{dt}\tilde{f}_T(\boldsymbol{x}, t) &= -\boldsymbol{k}^{(T)}(\boldsymbol{x})^\top \boldsymbol{r}_T(t), \\
\tilde{f}_T(\boldsymbol{x}, 0) &= f_m(\boldsymbol{x}, 0).
\end{aligned} \tag{30}$$

Let $g(\boldsymbol{x}, t) := f_m(\boldsymbol{x}, t) - \tilde{f}_T(\boldsymbol{x}, t)$. Then,

$$\begin{aligned}
\frac{d}{dt}g(\boldsymbol{x}, t) &= -\big(\boldsymbol{k}^{(t)}(\boldsymbol{x}) - \boldsymbol{k}^{(T)}(\boldsymbol{x})\big)^\top \boldsymbol{r}(t) - \boldsymbol{k}^{(T)}(\boldsymbol{x})^\top \boldsymbol{h}(t), \\
g(\boldsymbol{x}, 0) &= 0.
\end{aligned} \tag{31}$$

Integrating Equation 31 from 0 to $T$ and applying the triangle inequality and Cauchy-Schwarz inequality,

$$|g(\boldsymbol{x}, T)| \le \int_0^T \|\boldsymbol{k}^{(t)}(\boldsymbol{x}) - \boldsymbol{k}^{(T)}(\boldsymbol{x})\|\|\boldsymbol{r}(t)\|dt + \int_0^T \|\boldsymbol{k}^{(T)}(\boldsymbol{x})\|\|\boldsymbol{h}(t)\|dt. \tag{32}$$

By Assumption 3.4, $\|\boldsymbol{k}^{(t)}(\boldsymbol{x})\| \le \sqrt{K_m^{\max}}$ for all $t, \boldsymbol{x}$. Since $\boldsymbol{k}^{(t)}(\boldsymbol{x})$ consists of entries of $K_m^{(t)}$, it follows from the definition of the operator norm that

$$\|\boldsymbol{k}^{(t)}(\boldsymbol{x}) - \boldsymbol{k}^{(T)}(\boldsymbol{x})\| \le \|K_m^{(t)} - K_m^{(T)}\|. \tag{33}$$

Combining Equation 32, Equation 29, and Equation 33 yields

$$|g(\boldsymbol{x}, T)| \le \Big(1 + \sqrt{K_m^{\max}}\,T\Big)\Big(\sup_{u\in[0,T]} \|K_m^{(u)} - K_m^{(T)}\|\Big)\int_0^T \|\boldsymbol{r}(t)\|dt. \tag{34}$$

To control the residual integral, note that

$$
\begin{aligned}
\frac{d}{dt}\|\boldsymbol{r}(t)\|^2 &= 2\boldsymbol{r}(t)^\top \frac{d}{dt}\boldsymbol{r}(t) \\
&= -2\boldsymbol{r}(t)^\top K_m^{(t)} \boldsymbol{r}(t) \\
&\leq -2\,\mu_{\min}\big(K_m^{(t)}\big)\,\|\boldsymbol{r}(t)\|^2.
\end{aligned}
\tag{35}
$$

Under Assumption 3.3 and 3.5, there exists constant $C_r > 0$ that satisfies $\mu_{\min}\big(K_m^{(t)}\big) \geq C_r\,\mu_{\min}\big(K_m^{(0)}\big)$ for all $t \in [0,T]$. Thus, applying Grönwall's inequality to Equation 35 gives $\|\boldsymbol{r}(t)\| \leq e^{-C_r\,\mu_{\min}\big(K_m^{(0)}\big)t}\|\boldsymbol{r}(0)\|$. Therefore, the integral $\int_0^T \|\boldsymbol{r}(t)\|dt$ can be upper bounded as follows:

$$
\int_0^T \|\boldsymbol{r}(t)\|dt \leq \frac{1}{C_r\,\mu_{\min}\big(K_m^{(0)}\big)}.
\tag{36}
$$

Thus, Equation 34 is upper bounded by:

$$
\begin{aligned}
|g(\boldsymbol{x},t)| &= |f_m(\boldsymbol{x},t) - \tilde{f}_T(\boldsymbol{x},t)| \\
&\leq \Big(1 + \sqrt{K_m^{\max}}\,T\Big) \cdot \Big(\sup_{u\in[0,T]} \|K_m^{(u)} - K_m^{(T)}\|\Big) \cdot \frac{1}{C_r\,\mu_{\min}\big(K_m^{(0)}\big)}
\end{aligned}
\tag{37}
$$

Furthermore, the gradient flow with frozen kernel converges exponentially: $\boldsymbol{r}_T(t) = e^{-K_m^{(T)}(t-T)}\boldsymbol{r}_T(T)$ implies

$$
\begin{aligned}
|\widehat{f}_m^{(T)} - \tilde{f}_T(\cdot,T)| &\leq \sqrt{K_m^{\max}} \int_T^\infty \|\boldsymbol{r}_T(t)\|dt \\
&\leq \frac{\sqrt{K_m^{\max}}}{\mu_{\min}\big(K_m^{(T)}\big)}\|\boldsymbol{r}_T(T)\| \\
&\leq \frac{\sqrt{K_m^{\max}}}{\mu_{\min}\big(K_m^{(0)}\big)}\|\boldsymbol{r}_T(T)\|.
\end{aligned}
\tag{38}
$$

where we use Assumption 3.5 to compare $\mu_{\min}(K_m^{(T)})$ and $\mu_{\min}(K_m^{(0)})$. Combining Equation 37 and Equation 38 therefore yields

$$
|f_m(T) - \widehat{f}_m^{(T)}| \leq \frac{C_{\mathrm{lin}}}{\mu_{\min}\big(K_m^{(0)}\big)} \sup_{u\in[0,T]} \|K_m^{(u)} - K_m^{(T)}\|,
\tag{39}
$$

with $C_{\mathrm{lin}}$ independent of $\mu_{\min}$ and $m$.

**(G2) Time-variation of the empirical KRR predictor.** Using the ridgeless representation $\widehat{f}(\boldsymbol{x}) = \boldsymbol{k}_{\boldsymbol{x}}^\top K^{-1}\boldsymbol{y}$ and the resolvent identity $A^{-1} - B^{-1} = A^{-1}(B-A)B^{-1}$, together with $\|K^{-1}\| = 1/\mu_{\min}(K)$ and $\|\boldsymbol{k}_{\boldsymbol{x}}\| \leq \sqrt{K_m^{\max}}$, we obtain

$$
|\widehat{f}_m^{(T)} - \widehat{f}_m^{(0)}| \leq \frac{C_{\mathrm{tv}}}{\mu_{\min}^2\big(K_m^{(0)}\big)} \|K_m^{(T)} - K_m^{(0)}\|.
\tag{40}
$$

**(G3) Initialization vs. infinite width.** Applying the same resolvent argument to $K_m^{(0)}$ and $K_\infty$, and using Assumption 3.5 to compare their smallest eigenvalues via $\mu_{\min}(K_m^{(0)})$, one gets

$$
|\widehat{f}_m^{(0)} - f_\infty| \leq \frac{C_{\mathrm{init}}}{\mu_{\min}^2\big(K_m^{(0)}\big)} \|K_m^{(0)} - K_\infty\|.
\tag{41}
$$

**Synthesis of (G1)–(G3).** A simple row-sum bound using Assumption 3.4 implies $\mu_{\max}(A) \leq C_{\mathrm{spec}}$ for any relevant PSD matrix $A$, so $1/\mu_{\min}(A) \leq C_{\mathrm{spec}}/\mu_{\min}(A)^2$. This enables us to factor out $1/\mu_{\min}(K_m^{(0)})^2$. Combining Equation 39–Equation 41, we obtain

$$
|f_m(T) - f_\infty| \leq \frac{C_*}{\mu_{\min}^2\big(K_m^{(0)}\big)} \Big(\sup_{u\in[0,T]} \|K_m^{(u)} - K_m^{(T)}\| + \|K_m^{(T)} - K_m^{(0)}\| + \|K_m^{(0)} - K_\infty\|\Big).
\tag{42}
$$

**From functional difference to test error.** Define
$$E_g^{(m)} := \mathbb{E}_{(X,\boldsymbol{y})\sim\mathcal{D}}\big[(f(\boldsymbol{x}) - f_m(\boldsymbol{x},T))^2\big]$$
and $E_g^{(\infty)}$ analogously. A Cauchy–Schwarz inequality yields
$$E_g^{(m)} - E_g^{(\infty)} \le \sqrt{2}\,|f_m(T) - f_\infty|\,\big(E_g^{(m)} + E_g^{(\infty)}\big)^{1/2}.$$
Considering that the generalization errors are uniformly bounded by positive constants, substituting Equation 42 and using Theorem 3.2 to control $E_g^{(\infty)}$ produces Equation 8 and completes the proof of Theorem 3.7.

### A.4 PROOF OF THEOREM 3.8 (LAZY TRAINING)

Under Assumption 3.6,
$$\sup_{u\in[0,T]}\big\|K_m^{(u)} - K_m^{(0)}\big\| = \mathcal{O}\big(\phi(m)\big), \qquad \big\|K_m^{(0)} - K_\infty\big\| = \mathcal{O}\big(\phi(m)\big).$$
Hence
$$\|K_m^{(T)} - K_m^{(0)}\| \le \sup_u \|K_m^{(u)} - K_m^{(0)}\| = \mathcal{O}\big(\phi(m)\big),$$
and
$$\sup_u \|K_m^{(u)} - K_m^{(T)}\| \ \le \ \sup_u \|K_m^{(u)} - K_m^{(0)}\| + \|K_m^{(T)} - K_m^{(0)}\| \ = \ \mathcal{O}\big(\phi(m)\big).$$
Therefore,
$$\sup_u \|K_m^{(u)} - K_m^{(T)}\| + \|K_m^{(T)} - K_m^{(0)}\| + \|K_m^{(0)} - K_\infty\| \ = \ \mathcal{O}\big(\phi(m)\big).$$
Inserting this bound into Equation 42 yields
$$|f_m(T) - f_\infty| \ \le \ C\,\frac{\phi(m)}{\mu_{\min}\big(K_m^{(0)}\big)^2}.$$
Combining with the infinite-width control from Theorem 3.2 gives
$$E_g^{(m)} \ \le \ \frac{C_4}{\mu_{\min}\big(K_m^{(0)}\big)^2} \ + \ C_5\,\frac{\phi(m)}{\mu_{\min}\big(K_m^{(0)}\big)^2},$$
which is Equation 9 and completes the proof.

The three proofs above are controlled by a single quantity, the smallest eigenvalue. For the infinite-width predictor, $\mu_{\min}(G)$ controls both bias and variance through Equation 19. For finite width, $\mu_{\min}\big(K_m^{(0)}\big)$ governs the stability of kernel linearization, the time variation of the empirical KRR predictor, and the proximity to the infinite-width kernel. In the lazy regime, these effects incur only an $\phi(m)$ correction, smoothly connecting practice to the NTK theory.

## B VERIFICATION OF THE ASSUMPTIONS

In this section, we validate Assumption 3.5 and 3.6 on a two–layer ReLU regressor trained on synthetic datasets exactly as in Section 3. For each width $m$, we form the empirical NTK at initialization $K_m^{(0)}$ and after $T{=}1000$ epochs, $K_m^{(T)}$. We then evaluate the following quantities used in the assumptions:
$$\frac{\mu_{\min}\big(K_m^{(T)}\big)}{\mu_{\min}\big(K_m^{(0)}\big)}, \qquad \frac{\sup_{u\in[0,T]}\big\|K_m^{(u)} - K_m^{(0)}\big\|}{\big\|K_m^{(0)}\big\|}, \qquad \frac{\big\|K_m^{(0)} - K_\infty\big\|}{\big\|K_\infty\big\|}.$$
Figure 3 shows that the smallest eigenvalues at initialization and after training both increase with width and then saturate, approaching the infinite-width limit. Moreover, their ratio remains bounded away from 0 and $\infty$ across the sweep, indicating relative spectral stability. Figure 4 reports the relative kernel change: both the pathwise supremum $\sup_{u\in[0,T]} \|K_m^{(u)} - K_m^{(0)}\|$ and the terminal difference $\|K_m^{(0)} - K_\infty\|$ decrease as $m$ grows, supporting the assumption on the kernel change. The left plot shows $\sup_{u\in[0,T]} \|K_m^{(u)} - K_m^{(0)}\|$ follows $\mathcal{O}(m^{-1/2})$, which reproduces the result of Lee et al. (2019), while the right figure indicates the decrease of $\|K_m^{(0)} - K_\infty\|$ along width is far sharper. In either case, these kernel changes would be upper bounded by $\phi(m)$. These plots support the assumptions used in our finite-width bounds.

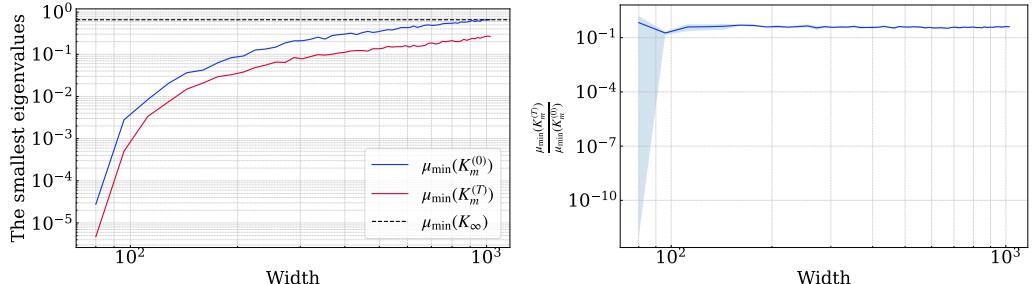

Figure 3: **Smallest NTK eigenvalue vs. width (two–layer ReLU).** Left: comparison of $\mu_{\min}\big(K_m^{(0)}\big)$, $\mu_{\min}\big(K_m^{(T)}\big)$, and the infinite–width limit $\mu_{\min}(K_\infty)$. Right: spectral ratio of these eigenvalues. Both eigenvalue curves increase with width and then saturate toward $\mu_{\min}(K_\infty)$, while $\rho_m$ remains bounded away from 0 and $\infty$, supporting relative spectral stability of Assumption 3.5.

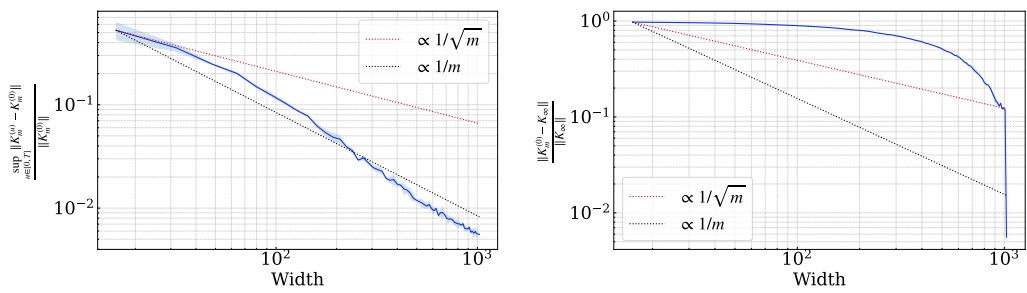

Figure 4: **Relative kernel change (two-layer ReLU).** Pathwise drift $\sup_{u\in[0,T]}\|K_m^{(u)} - K_m^{(0)}\|$ and initialization drift $\|K_m^{(0)} - K_\infty\|$ both decrease as width grows, consistent with a width–dependent decay $\phi(m)\downarrow 0$ and supporting Assumption 3.6.

## C    EXPERIMENTAL SETTINGS

### C.1    TRAINING

**The case of DNN:** We construct a 10-layer fully connected ReLU network, where the width is consistent among layers. The width $m$ is swept over $m \in \{4, 8, \ldots, 512\}$. The network is trained on the Diabetes Dataset and the California Dataset. Each dataset is divided into 80% for training and 20% for test samples. Models are initialized with He initialization (He et al., 2015) and optimized with Adam (learning rate $10^{-3}$), mini–batch size 32, for 500 epochs. Unless otherwise stated, we report test performance from the final checkpoint.

**The case of CNN:** The convolutional baseline is a small two-stage network with $3\times3$ convolutions, batch normalization, and ReLU, followed by global average pooling and a linear output head. Spatial resolution is preserved inside the convolutional stack and reduced only by the global pooling. We treat the channel count $c$ as the width parameter and sweep $c \in \{4, 8, \ldots, 128\}$ while keeping depth and all other components fixed. Training uses the same protocol as the DNN: Adam with learning rate $10^{-3}$, batch size 32, for 500 epochs, with He initialization. During training, batch normalization uses batch statistics; during evaluation, it uses fixed stored statistics. We report test performance at the final epoch. The CNNs are trained on CIFAR-10 and MNIST. Each dataset is divided into 80% for training and 20% for test samples. The models are trained with the same hyperparameters as DNNs.

**The case of ResNet:** The residual baseline uses a $3\times3$ stem followed by a fixed number of residual blocks. Each block has two $3\times3$ convolutions with batch normalization and ReLU, and an identity skip connection. A global average pooling layer feeds a linear head with a single output unit. We control width through the base channel count $c$ and sweep $c \in \{4, 8, \ldots, 128\}$, holding the number of blocks constant. Optimization and initialization follow the same settings as above (Adam $10^{-3}$, batch size 32, 500 epochs, He initialization). Batch normalization uses batch statistics for training and stored statistics for evaluation. Results are taken from the final checkpoint. CIFAR-10 and MNIST are also employed for training these ResNets. Additionally, we train the models with the same hyperparameters as DNN and CNN.

Each network is trained on an NVIDIA 40GB A100 GPU.

## C.2 ALGORITHM 1

We follow Algorithm 1 to compute the empirical NTK at initialization and to estimate its spectral edge. Architectures, datasets, width grids, and initialization match the training setup in Sec. 3.1: for fully connected networks, we sweep $m \in \{4, 8, \ldots, 512\}$; for convolutional and residual networks, we sweep channel counts $\{4, 8, \ldots, 128\}$ while keeping depth fixed. Labels are not used. Batch normalization is evaluated in inference mode when forming the kernel to avoid dependence on batch statistics.

For each width $m$, we construct the empirical kernel $K_m^{(0)}$ at initialization and estimate its smallest eigenvalue using the LOBPCG solver with a tolerance $10^{-8}$ and at most 500 iterations. Denote the resulting estimate by $\widehat{\mu}_{\min}(m)$. We then fit a smooth parametric curve $g(m; \vartheta)$ to $\{(m, \widehat{\mu}_{\min}(m))\}$ by least squares and determine the cardinal width with $\delta = 0.01$.

# D NOTES FOR THE CARDINAL WIDTH

## D.1 INSENSITIVITY OF THE CARDINAL WIDTH TO TRAINING RECIPES

We assess the robustness of the cardinal width on a DNN regressor (UCI Diabetes) under the same setup as Section 4, varying the optimizer, learning rate, and initialization methods: Adam with learning rates $10^{-3}$ and $5\times10^{-4}$ for 500 epochs, and SGD with step–decay schedules $0.1/\sqrt{t}$ and $0.2/\sqrt{t}$ for 1000 epochs, where $t$ denotes the number of steps. The networks are initialized with He initialization and NTK initialization. The batch size is 32 throughout.

Figure 5 shows that the test loss rapidly drops for small widths followed by a flat region beginning at widths on the order of $10^2$ for all the settings. The location of the cardinal width varies little across recipes. For Adam (both learning rates, both He and NTK initializations), the cardinal width concentrates in a narrow band around 100. For SGD, it is slightly to the right but remains within the same order, and the plateau level shifts modestly without materially moving the cardinal width. Naturally, under extreme hyperparameters, including very large learning rates, the cardinal width can shift. However, within the practically reasonable settings used in Figure 5, it remains within a narrow range. In short, the predicted saturation width is broadly stable over practical optimizer and learning–rate choices, consistent with Note 2 in the main paper.

## D.2 SUBSAMPLING IN NTK COMPUTATION

We repeat the experiments of Section 4, computing $\mu_{\min}\big(K_m^{(0)}\big)$ on the full training set and on random subsets while holding all other settings fixed. For each subsample rate, we sweep the width $m$, fit $g(m; \vartheta)$, and estimate the cardinal width via Algorithm 1. Figure 6 summarizes the outcomes. On Diabetes dataset, the full-data estimate is 94.40, and rates 0.10 and 0.50 give 90.58 ($-4.0\%$) and 67.66 ($-28.3\%$). On California Housing dataset, the full-data estimate is $m_{\text{card}} \approx 114.77$, and subsample rates 0.10 and 0.50 yield 89.30 ($-22.1\%$) and 98.22 ($-14.4\%$), respectively. Note that the estimated width in full sampling differs from the result of Figure 2, due to the randomness of the fitting curve. In both datasets, the cardinal width remains reasonably stable under 10–50% subsampling, indicating that training-free width selection can be performed on a reduced fraction of inputs with only modest loss in accuracy and with computation roughly proportional to the subsample rate.

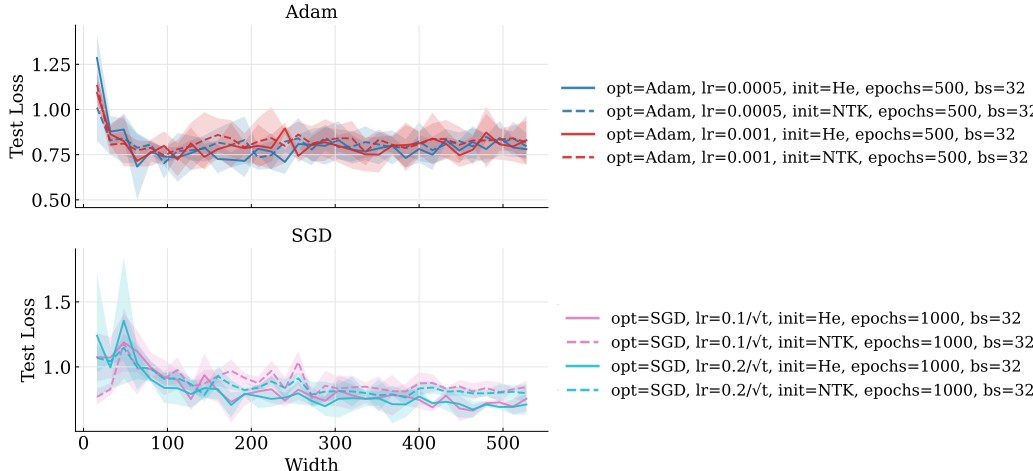

Figure 5: **Cardinal width is stable across training recipes (DNN on Diabetes).** Test loss versus width for Adam (top) and SGD (bottom) with two learning rates/schedules and two initializations (He, NTK). All curves exhibit a sharp drop followed by a plateau beginning at widths $\sim 100$, with only minor shifts in the cardinal width across recipes.

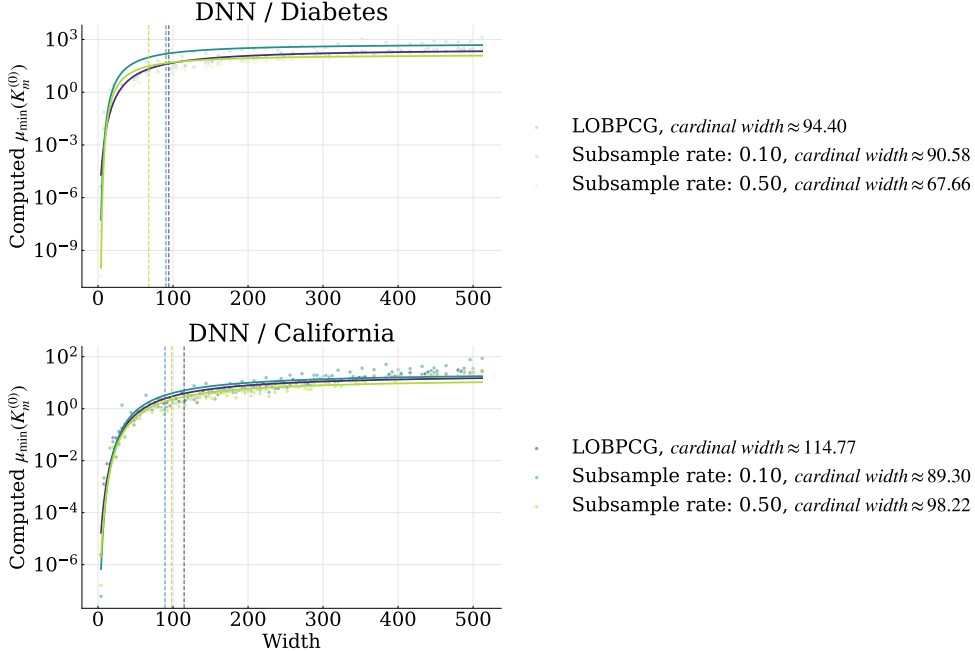

Figure 6: **Subsampling preserves the cardinal width.** Computed $\mu_{\min}\big(K_m^{(0)}\big)$ versus width for DNNs. The curves are fits of $g(m;\vartheta)$ and vertical dashed lines mark the estimated cardinal widths. The predicted cardinal widths are broadly stable under 10–50% subsampling, supporting a lower-cost, training-free width selection.

## E EXTENSION TO CLASSIFICATION

While our theory focuses on regression tasks following previous NTK-based works, we now discuss the potential extension of our method to classification tasks. We apply our Algorithm 1 to determine

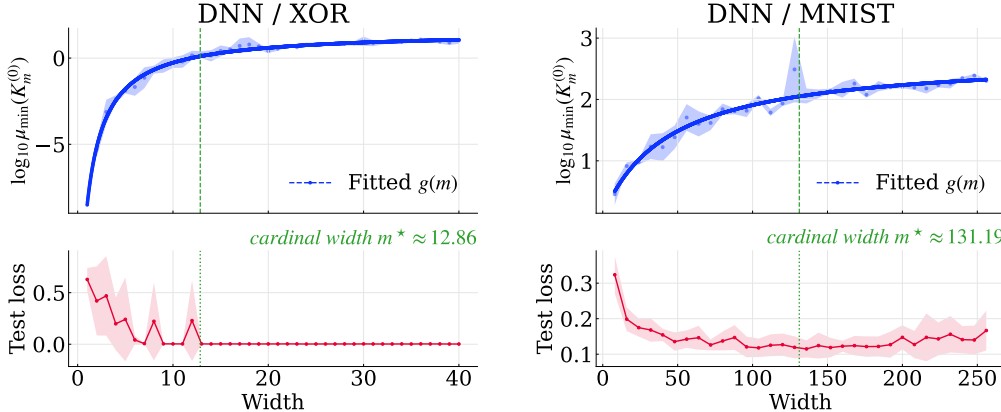

Figure 7: **The predicted cardinal width in classification tasks.** At the cardinal width identified using our algorithm, the test loss correspondingly saturates, indicating the potential extension of our method to classification tasks.

the cardinal width for 10-layer DNNs trained to solve the classification task on the stylized XOR data introduced in (Wei et al., 2019) and MNIST. We prepare XOR data with 2000 samples for training and 100 for testing. The input dimension is set to 20.

In our experiment, the network is optimized via Adam with a learning rate $10^{-3}$, and trained for 20 epochs on the XOR data and 50 epochs on MNIST. The batch size is 32 for both datasets. Figure 7 shows that the estimated cardinal width is aligned with the width at which the test loss no longer improves for both the XOR data and MNIST. These results imply that our algorithm can potentially be applied to classification problems, although it is not theoretically guaranteed.

## F  LLM USAGE

We used a large language model tool solely for editing during manuscript preparation, such as grammar corrections, wording clarification, and minor condensation of prose. All the text suggested by the tool was reviewed and revised by the authors. All ideas, methods, experiments, and conclusions are developed by the authors.

