# OpenReview forum: "Training-Free Determination of Network Width via Neural Tangent Kernel"
_ICLR.cc/2026/Conference — ICLR 2026 Poster_

### Official Review · Reviewer_khFg · 2025-10-28

**Soundness:** 2
**Presentation:** 3
**Contribution:** 3
**Rating:** 6
**Confidence:** 3

**Summary:**

The paper shows that the test error of infinite and finite-width networks is upper bound by a value related to the smallest eigenvalue of the (empirical) NTK. Based on the theory, the authors provide a training-free method to determine the cardinal width, which is the critical width where the test loss saturates, through examining the saturation of the empirical NTK smallest eigenvalue at initialization. Experiments show that the proposed method can predict cardinal width pretty well.

**Strengths:**

1. There are many empirical works trying to utilize the empirical NTK spectrum to study the neural network properties at initialization. The theory derived in the paper offers a principled way of choosing the proper width without model training, which could be useful for practitioners in the field.
2. The paper is well-organized. The theory is accompanied with proper experiments.

**Weaknesses:**

1. The main concern of the paper is that whether the cardinal width is useful in practice. In several plots of figure 2, in CNN/MNIST from width 50 to 125 I do not observe obvious slope change. Similar observations can be seen in CNN/CIFAR10 width 50-100 and  ResNet/MNIST width 25-75. Hence, it makes the “plateau point” ambiguous. In such cases, a coarse manual random guess could possibly achieve similar conclusions. In conclusion, I am unsure about the superiority of the proposed method compared to heuristic rule.
2. Based on the figure 1, the observed plot is closer to $1/\sqrt{\mu_{min}}$, raising the concern of whether the bound is loose in real settings. Could the authors provide a detailed comparison/discussion with existing theoretical results regarding the NTK-based test error bound?

**Questions:**

In scaling law studies (Kaplan et al. 2020), the authors show that width and depth have minimal effects within a wide range compared to overall parameter count. Can the authors provide more insights on the superiority of the proposed methods in practical uses in more details, for example, how can the proposed method be used in real world experiment settings, like scaling law experiments.

---

> ### Author Response · Authors · 2025-11-21
> **We thank Reviewer khFg for reviewing our work**
>
> We thank the reviewer for the overall positive feedback on our work. We appreciate your recognition that our algorithm without training is useful and the paper is well-structured.
> We are happy to clarify any further concerns or questions below.
>
> **Weakness 1:**
> > The main concern of the paper is that whether the cardinal width is useful in practice. In several plots of figure 2, in CNN/MNIST from width 50 to 125 I do not observe obvious slope change. Similar observations can be seen in CNN/CIFAR10 width 50-100 and ResNet/MNIST width 25-75. Hence, it makes the ``plateau point'' ambiguous. In such cases, a coarse manual random guess could possibly achieve similar conclusions. In conclusion, I am unsure about the superiority of the proposed method compared to heuristic rule.
>
> Thank you for your comment. We agree with Reviewer khFg that the test loss improves slightly around "cardinal width" in some settings, i.e. the exact saturation point can be ambiguous. This ambiguity however is a characteristic of the loss curves, rather than something specific to our method, and any heuristic or NAS will face the same difficulty with the ambiguity. In this context, nonetheless, the proposed method is rather deterministic, as justified by our generalization bounds based on the smallest eigenvalue of NTK.
>
> The experimental results also show that cardinal width is indeed useful. In fact, only slight improvement of test loss is observed beyond cardinal width, suggesting that further increases in width bring limited benefit. Additionally, our method is superior to the heuristic one in clear saturation cases such as DNN/Diabetes and DNN/California. Furthermore, our method does not require training unlike heuristic rules (such as naive grid search), which is an advantage in terms of efficiency as well as clarity.
>
> ---
>
> **Weakness 2**
> > Based on the figure 1, the observed plot is closer to $1 / \sqrt{\mu_{\min}}$
> , raising the concern of whether the bound is loose in real settings. Could the authors provide a detailed comparison/discussion with existing theoretical results regarding the NTK-based test error bound?
>
> We thank the reviewer for raising this point. We agree that in Figure 1 the empirical scaling looks closer to $1/\sqrt{\mu_{\min}}$ than to our theoretical $1/\mu_{\min}^2$ bound; this is expected, because our result comes from replacing the full spectral risk expression with a worst-case inequality that depends only on $\mu_{\min}$, which necessarily sacrifices tightness.
>
> By contrast, NTK-based works such as [1] [2], and [3] describe test error as (approximate) equalities using entire eigenvalues. These are more precise for designing the error bound, but they are not designed as computationally cheap and practical rules for choosing width.
>
> Our goal is complementary. We deliberately trade tightness for computational efficiency by using $\mu_{\min}$ alone, which directly controls the error bound and allows us to define a cardinal width at the saturation of $\mu_{\min}$ with practical computational time. Even if the bound is not tight, the predicted cardinal width aligns with the width where the test loss saturates, ensuring the effectiveness of the bound.
>
> ---
>
> **Question 1**
> > In scaling law studies (Kaplan et al. 2020), the authors show that width and depth have minimal effects within a wide range compared to overall parameter count. Can the authors provide more insights on the superiority of the proposed methods in practical uses in more details, for example, how can the proposed method be used in real world experiment settings, like scaling law experiments.
>
> Thank you for your comment. We would like to argue that the setting of this paper is different from that of the scaling law.
>
> In the scaling law study (Kaplan et al., 2020), the main observation is that for a fixed number of non-embedding parameters $N$, changing the model width has only a mild effect on performance. This is a statement about shape invariance at **fixed $N$**.
>
> By contrast, in our experiments we vary width at fixed depth, so changing width also changes $N$ accordingly. In fact, Figure 2 in our paper shows the significant improvement of the test loss with changing width. Thus, the proposed method is useful in real world experiment settings, when the number of parameters accordingly changes with changing width (which are very general in scaling models).
>
>
> [1] Blake Bordelon et al. Spectrum dependent learning curves in kernel regression and wide neural networks. ICML 2020.
>
> [2] Abdulkadir Canatar et al. Spectral bias and task-model alignment explain generalization in kernel regression and infinitely wide neural networks. Nature Communications 2021.
>
> [3] James B. Simon et al. The eigenlearning framework: a conservation law perspective on kernel ridge regression and wide neural networks. TMLR 2023.

---

### Official Review · Reviewer_fdRP · 2025-10-28

**Soundness:** 2
**Presentation:** 3
**Contribution:** 2
**Rating:** 6
**Confidence:** 3

**Summary:**

The paper studies generalization loss of neural nets in the kernel regime and derives a generalization bound that depends on the minimum eigenvalue of the kernel at initialization ($\mu_{min}$). In particular their results indicate an upperbound proportional to $1/\mu_{min}^2$ for the test loss. Moreover, the result holds for both infinite and finite width neural networks. This leads to an algorithm for a training-free selection for the network width where the point where increasing the width does not significantly help with increasing $\mu_{min}$ is selected as the best choice for width.

**Strengths:**

Rigorous theory and several experiments to verify the effectiveness of the proposed algorithm and theory. Bounding the test loss based on a single computable parameter can be insightful even in the restricted setup of NTK regime and gradient flow considered in this paper. The authors also present a simple algorithm for width selection which is computationally efficient and does not require training.

**Weaknesses:**

-The paper mentions the importance of kernel condition number for both optimization and generalization as specified in literature. Can the authors please explain more the relation to prior works in line 90 of the paper? Moreover, the authors claim that they are the first to show the relation between $\mu_{min}$ and test error in the kernel regime. In my opinion, it needs more clarification by authors as there is already a link between these two in the literature. In particular, it is known that the neural network objectives of sufficiently large width with square-loss satisfy a PL condition in the kernel regime (e.g, Thm 4. in [1] where the authors show the relation between PL parameter and the minimum eigenvalue of kernel) and therefore one can obtain test loss bounds based on $\mu_{min}$ (e.g., see [2]). The minimum eigenvalue also seems to be proportional to class margin for classification tasks as discussed by [3] (section 5).

-How does $\mu_{min}$ scale with number of samples ($n$)? Also, It's better to clarify whether $C_1,C_2$ in theorem 3.2. scale with $n$. can the authors please clarify how the bound scales with $n$, or specify relevant work in literature?

-What is the benefit of bounding test loss only based on $\mu_{min}$ beyond computational efficiency reasons? can including more information beyond the minimum eigenvalue (while not using the whole spectrum) help with generalization bounds? I seems the bounds based only on $\mu_{min}$ can be loose (or at least the authors do not characterize tightness) and moreover the authors observe a tighter $1/\sqrt{\mu_{min}}$ relation in their experiments.

-Do the authors suspect that the theory can be extended to the feature-learning regime using DMFT framework of [Bordelon and Pehlevan, 2023] and the resulting equations to specify whether there is relation between generalization gap and $\mu_{min}$ of dynamical (time-varying) kernel? Has such a relation been observed in experiments?

-Can the authors verify their theory on stylized data (e.g. boolean XOR data or multi-index models (see references[3-4])) to see how their theory predicts compared to current theoretical lower bounds for the network width in the kernel regime?

-Does Thm 3.8 hold for any $m$? or Should the network be sufficiently wide for this results to hold? It seems to me that this detail is not emphasized in the statement of the theorem.

1- Loss landscapes and optimization in over-parameterized non-linear systems and neural networks, Liu et al, 2021

2- Sharper Generalization Bounds for Learning with Gradient-dominated Objective Functions, Lei and Ying, ICLR 2021.

3- Polylogarithmic width suffices for gradient descent to achieve arbitrarily small test error with shallow relu networks. Ji and Telgarsky, ICLR 2020.

4- Feature selection and low test error in shallow low-rotation relu networks, Telgarsky, ICLR 2023.

**Questions:**

Please see above section for questions.

---

> ### Author Response · Authors · 2025-11-21
> **We thank Reviewer fdRP for reviewing our work (1/3)**
>
> We thank the reviewer for the overall positive feedback on our work. We appreciate your recognition of our theoretical rigor and algorithmic efficiency.
> We are happy to clarify any further concerns or questions below.
>
> ---
>
> > The paper mentions the importance of kernel condition number for both optimization and generalization as specified in literature. Can the authors please explain more the relation to prior works in line 90 of the paper? Moreover, the authors claim that they are the first to show the relation between $\mu_{\min}$ and test error in the kernel regime. In my opinion, it needs more clarification by authors as there is already a link between these two in the literature. In particular, it is known that the neural network objectives of sufficiently large width with square-loss satisfy a PL condition in the kernel regime (e.g, Thm 4. in [1] where the authors show the relation between PL parameter and the minimum eigenvalue of kernel) and therefore one can obtain test loss bounds based on $\mu_{\min}$ (e.g., see [2]). The minimum eigenvalue also seems to be proportional to class margin for classification tasks as discussed by [3] (section 5).
>
> Thank you for your comment.
> First, we fully agree that the smallest eigenvalue $\mu_{\min}$ is certainly an important quantity in the NTK field, but the clear and direct connection between $\mu_{\min}$ and test error has not been stated, to the best of our knowledge.
>
> [Nguyen et al. Tight bounds on the smallest eigenvalue of the neural tangent kernel for deep relu networks. ICML 2021] and [Karhadkar et al. Bounds for the smallest eigenvalue of the NTK for arbitrary spherical data of arbitrary dimension. NeurIPS 2024] derive lower bounds on the smallest eigenvalue, which in turn support global convergence guarantees in works such as [Du et al. Gradient descent provably optimizes over-parameterized neural networks. ICLR 2019] and [Montanari and Zhong. The interpolation phase transition in neural networks: Memorization and generalization under lazy training. Annals of Statistics 2022].
>
> In the kernel regression context, there is also a rich body of work characterizing generalization via the full eigenspectrum [Canatar et al. Spectral bias and task-model alignment explain generalization in kernel regression and infinitely wide neural networks. Nature Communications 2021], [Simon et al. The eigenlearning framework: a conservation law perspective on kernel ridge regression and wide neural networks. TMLR 2023], etc. However, to the best of our knowledge, none of these works directly tie generalization error solely to the smallest eigenvalue of NTK. Our theoretical novelty is precisely
> -  to provide a direct connection between the smallest eigenvalue of the NTK and generalization error
> -  to avoid dependence on the full spectrum so that the resulting bounds can be efficiently used for practical width selection.
>
> Next, let us address the Reviewer fdRP's concerns about our novelty compared to [1] (Liu et al., ACHA2021), [2] (Lei and Ying, ICLR2021), and [3] (Ji and Telgarsky, ICLR 2020). First, combining [1] and [2] does not yield our results. The bounds in [2] are PL-based and therefore **algorithm-dependent**. They explicitly depend on the convergence speed and hyperparameters of the learning algorithm (step sizes, number of iterations, stability conditions). Even if we inject the relation between the PL constant $\beta$ and the NTK’s smallest eigenvalue $\mu_{\min}$ of [1] into the generalization bounds of [2], the resulting bounds remain dependent on the optimization dynamics, while our theorems provide **algorithm-independent** upper bounds that depend on $\mu_{\min}$. Moreover, the combination of [1] and [2] is itself non-trivial and, to the best of our knowledge, has not been worked out explicitly in the literature. Our work is, to our knowledge, the first to state and analyze a direct relationship between $\mu_{\min}$ and generalization error in  finite width networks, and we furthermore leverage this relationship to design a practical width selection rule.
>
> Second, [3] (Ji and Telgarsky, ICLR 2020) study separability in classification via the NTK margin for infinite-width, two-layer networks. In contrast, Theorem 3.8 holds at finite width $m$ (although sufficiently large) and extends to networks of any depth by using the results of [Lee et al., NeurIPS 2019]. Thus, our setting and scope are different and in this sense strictly more general.

---

> ### Author Response · Authors · 2025-11-21
> **We thank Reviewer fdRP for reviewing our work (2/3)**
>
> > How does $\mu_{\min}$ scale with number of samples ($n$)? Also, It's better to clarify whether $C_1, C_2$ in theorem 3.2. scale with $n$. can the authors please clarify how the bound scales with $n$, or specify relevant work in literature?
>
> First, the scale of $\mu_{\min}$ with respect to sample $n$ is provided in [Karhadkar et al. NeurIPS2024]. The authors show in Corollary 2 of their paper that $\mu_{\min}$ decays toward zero as the number of samples $n$ increases:
> $\mu_{\min} = \tilde{\Omega}(n^{-4/(d_0-1)})$ and $\mu_{\min} = \tilde{\mathcal{O}}(n^{-2/(d_0-1)})$ in a shallow ReLU network where $d_0$ denotes the input dimension and the hidden layer width $d_1$ is sufficiently wide.
>
> As for $C_1$ and $C_2$, these values are independent of $n$, using the result of [Canatar et al. Nature Communications 2021]. However, the number of samples $n$ can be viewed as constant in determining "cardinal width", since in our settings the dataset is already provided and thus $n$ does not change.
>
> ---
>
> > What is the benefit of bounding test loss only based on $\mu_{\min}$ beyond computational efficiency reasons? can including more information beyond the minimum eigenvalue (while not using the whole spectrum) help with generalization bounds? I seems the bounds based only on $\mu_{\min}$ can be loose (or at least the authors do not characterize tightness) and moreover the authors observe a tighter $1 / \mu_{\min}$ relation in their experiments.
>
> The benefit of using only $\mu_{\min}$ is primarily computational. We agree that using more of the spectrum could in principle give tighter bounds. Indeed, [Canatar et al. Nature Communications 2021] provide the exact equality of generalization error using whole eigenvalues in infinite-width networks. However, computing the whole eigenvalues rely on full eigendecompositions, which in the naive form cost $\mathcal{O}(n^3)$ in the number of samples and quickly become impractical in the real-world dataset.
>
> At the same time, $\mu_{\min}$ is exactly the quantity we need for our purpose. It controls the upper bound of generalization error, so our theory using $\mu_{\min}$ already tells us when the network is wide enough. Therefore, for choosing the cardinal width, using only $\mu_{\min}$ is sufficient in practice, as our experiments suggest.
>
> ---
>
> > Do the authors suspect that the theory can be extended to the feature-learning regime using DMFT framework of [Bordelon and Pehlevan, 2023] and the resulting equations to specify whether there is relation between generalization gap and $\mu_{\min}$ of dynamical (time-varying) kernel? Has such a relation been observed in experiments?
>
> [Reviewer thWr raised a similar point, so we repeat our unified response here.]
>
> Thank you for your insightful comment. Yes, the theory can be extended to the feature-learning regime using [Bordelon and Pehlevan, NeurIPS2023].
>
> [Bordelon and Pehlevan, NeurIPS2023] show that in the Dynamical Mean Field Theory (DMFT), the fluctuations over random initialization of NTK are of order $\mathcal{O}(m^{-1/2})$, and the variance is $\mathcal{O}(1/m)$ for a finite width $m$ even in the feature-learning regime. The authors show that richer feature learning leads to closer agreement with infinite-width mean field behavior. Thus, the kernel change $\|K_m^{(T)} - K_m^{(0)}\|$ would converge to a constant (not necessarily zero as in lazy training). Theorem 3.7 in our paper shows the upper bound of the generalization error:
>
> $E_g^{(m)}
> \\le\
> E_g^{(\infty)}
> \+\ C_3\\frac{\sup_{u}\|K_m^{(u)}-K_m^{(0)}\|+\|K_m^{(T)}-K_m^{(0)}\|+\|K_m^{(0)}-K_\infty\|}{\mu_{\min}^2(K_m^{(0)})}.$
>
> We assume lazy training to ensure $(\sup_{u}\|K_m^{(u)}-K_m^{(0)}\|+\|K_m^{(T)}-K_m^{(0)}\|+\|K_m^{(0)}-K_\infty\|) \to 0$ in sufficiently wide networks and derive Theorem 3.8. However, even in the feature-learning regime, this part becomes a small constant in sufficiently wide networks if we apply the result of [Bordelon and Pehlevan, NeurIPS2023]. Therefore, the main message of our theory that **"The smallest eigenvalue of NTK controls the upper bound of generalization error"** remains valid in feature learning. Thus, $\mu_{\min}$ works as an indicator for "cardinal width" in both lazy-training and feature-learning regimes.
> In fact, the experiment in Figure 2 does not assume lazy training, but still the estimated cardinal width is aligned with the width where the test loss saturates.

---

> ### Author Response · Authors · 2025-11-21
> **We thank Reviewer fdRP for reviewing our work (3/3)**
>
> > Can the authors verify their theory on stylized data (e.g. boolean XOR data or multi-index models (see references[3-4])) to see how their theory predicts compared to current theoretical lower bounds for the network width in the kernel regime?
>
> Thank you for this suggestion. Our theory is developed for regression with squared loss and is not tailored to stylized classification data such as XOR data or multi-index models, so it does not directly apply to the upper bound for those settings.
>
> Within the scope of our assumptions, we have already verified the theory on synthesized regression data (Figure 1). Following your suggestion, we additionally checked whether our algorithm remains valid on stylized XOR data. In the revised manuscript, **Appendix E** reports experiments on XOR data (and, as an extension, on MNIST), showing that the predicted "cardinal width" still matches the width at which test loss saturates. The revised parts are highlighted in blue for your convenience.
>
> ---
>
> > Does Thm 3.8 hold for any $m$? or Should the network be sufficiently wide for this results to hold? It seems to me that this detail is not emphasized in the statement of the theorem.
>
> Thank you for your insightful comment. Theorem 3.8 implicitly holds for sufficiently large $m$, because the theory is based on the assumption of lazy training. This is noted just after Theorem 3.8 which says:
>
> *This is a special case of Theorem 3.7, which applies Assumption 3.6 representing the kernel change during training with the width $m$ in sufficiently wide networks.*

---

> > ### Comment · Reviewer_fdRP · 2025-11-23
> >
> > Thank you for your response and the additional numerical experiments.
> >
> > 1- The authors mentioned that their approach leads to algorithmic-independent bounds. How is this an advantage over previous bounds given that the PL condition already leads to global convergence and near-zero training error and in the NTK regime in general the NN objective behaves convex-like? Moreover, the authors mention combination of [1] and [2] is not trivial. This part is not really clear to me. In what sense are the authors claiming that the combination is non-trivial?
> >
> > In general, It would help the paper's clarity if these points are discussed in the paper.
> >
> > 2- It is unclear how large should the width be, for theorems to be valid. Many previous works have established bounds that require extremely large width for the NTK regime, e.g. [3] requires at least $d^8$ neurons for a boolean XOR data. This is in contrast to the experiments in this paper where the width is considerably smaller. Therefore, I'm still wondering how the theorems connect to the experiments. If the authors claim that their results apply to the feature-learning regime, this should be stated as a theorem in the paper.

---

> > > ### Author Response · Authors · 2025-11-30
> > > **Official Comment by Authors**
> > >
> > > > The authors mentioned that their approach leads to algorithmic-independent bounds. How is this an advantage over previous bounds given that the PL condition already leads to global convergence and near-zero training error and in the NTK regime in general the NN objective behaves convex-like? Moreover, the authors mention combination of [1] and [2] is not trivial. This part is not really clear to me. In what sense are the authors claiming that the combination is non-trivial? In general, It would help the paper's clarity if these points are discussed in the paper.
> > >
> > > Thank you for your insightful comment. Certainly, [2] can reduce to algorithm-independent if local PL condition assumed by [1] holds in Theorem 1 of [2] (although not stated explicitly in [2]).
> > >
> > > However, it should be noted that the combination is non-trivial, because there is a gap between the assumptions in [1] and [2]. For example, [1] assumes local PL condition in a ball around the initialization (lazy regime), but [2] assumes PL condition for all the parameter space based on the training dataset ([2] implies that PL condition can be relaxed to a local PL condition, but does not provide proof). Additionally, the assumption $L \le \frac{n \beta}{4}$ ($L$: $L$-smooth constant) in [2] does not always hold when we apply [1] to [2]. Translating the local PL constant in [1], which is controlled by the smallest eigenvalue of NTK $\mu_{\min}$, into the PL constant $\beta$ in [2] makes the smoothness condition: $L \le \frac{\mu_{\min}}{4}$. This is, for instance, not satisfied in linear models with least square loss, where $L$ equals to the largest eigenvalue of NTK.
> > >
> > > In this sence, we mentioned that the combination is non-trivial. Although a direct comparison is therefore not fully justified, it is stil instructive to note that the combination of [1] and [2] leads to the bound $\mathcal{O}(1/\mu_{\min})$, while our results is $\mathcal{O}(1/\mu^2_{\min})$. Whether the bound is tighter depends on the regime of $\mu_{\min}$.
> > >
> > > Thank you again for your suggestion and we added the discussion above after Theorem 3.8 in the main paper, about the novelty/clarity from the previous works such as [1] and [2]. We highlighted this additional discussion in red.
> > >
> > > >  It is unclear how large should the width be, for theorems to be valid. Many previous works have established bounds that require extremely large width for the NTK regime, e.g. [3] requires at least $d^8$ neurons for a boolean XOR data. This is in contrast to the experiments in this paper where the width is considerably smaller. Therefore, I'm still wondering how the theorems connect to the experiments. If the authors claim that their results apply to the feature-learning regime, this should be stated as a theorem in the paper.
> > >
> > > We agree that analysis of NTK often requires extremely large width, but our theorems are formulated instead in terms of kernel stability, rather than explicit lower bounds on width. For our goal of estimating cardinal width, the theorem works once kernel change becomes modest and the behavior is largely governed by $\mu_{\min}$, which may occur at comparatively moderate widths we observe in the experiments.

---

### Official Review · Reviewer_M1eP · 2025-11-01

**Soundness:** 3
**Presentation:** 3
**Contribution:** 2
**Rating:** 6
**Confidence:** 2

**Summary:**

This paper proposes a training-free criterion for selecting an appropriate network width. The key observation is that, when the width is sufficiently large, test error is governed by the smallest eigenvalue of the NTK and authors justify this both theoretically and empirically. Based on this, authors introduce a method that estimates the cardinal width, the point at which generalization performance saturates, by tracking the smallest NTK eigenvalue at initialization through Locally Optimal Block Preconditioned Conjugate Gradient . Experiments across multiple architectures and datasets validate effectiveness of this method.

**Strengths:**

The paper first makes an non trivial claim, test error is controlled by the smallest NTK eigenvalue, and supports it both theoretically and empirically. Leveraging this insight and pointing out the highlighting of using the smallest NTK eigenvalue rather than the full spectrum, the authors devise a practical procedure to estimate the cardinal width and experimentally validate the approach on two-layer networks. This line of research is compelling and provides well-balanced results.

**Weaknesses:**

Assumption 3.6, invariance of NTK, is central to the NTK framework, yet the authors assume it without proof. Although the empirical evidence is supportive, a full theoretical justification would substantially strengthen the work.

Although the effectiveness of the cardinal width is demonstrated experimentally, I’m concerned about how closely the experimental setup reflects real-world training conditions.

**Questions:**

Considering the eigenvalues of the NTK is a nice approach, but eigenvalues are sensitive quantities in numerical analysis. Did the authors report any uncertainty in calculating this eigenvalue? If not, do you have insights on how to compute it robustly in this setup?

In the EXPERIMENTAL VERIFICATION subsection, the weights seem to be initialized as ( \mathcal{N}(0,1) ), which appears different from NTK initialization. Could the authors explain this point?

**Details Of Ethics Concerns:**

There is no ethics concerns.

---

> ### Author Response · Authors · 2025-11-21
> **We thank Reviewer M1eP for reviewing our work**
>
> We thank the reviewer for the overall positive feedback on our work. We appreciate your recognition that our theory is non-trivial and the algorithm to determine width is practical.
> We are happy to clarify any further concerns or questions below.
>
> ---
>
> > Assumption 3.6, invariance of NTK, is central to the NTK framework, yet the authors assume it without proof. Although the empirical evidence is supportive, a full theoretical justification would substantially strengthen the work.
>
> Thank you for your comment. As Reviewer M1eP notes, we use Assumption 3.6 (invariance of NTK) without proof. However, the theoretical justification of the assumption is provided in [Lee et al. NeurIPS 2019]. The authors show in Theorem 2.1 in their paper that the invariance of NTK $\|K_m^{(T)} - K_m^{(0)} \|$ follows $\mathcal{O}(m^{-1/2})$ in DNNs in the lazy-regime. We have briefly provided the notation after Assumption 3.6 in the main paper.
>
> ---
>
> > Although the effectiveness of the cardinal width is demonstrated experimentally, I’m concerned about how closely the experimental setup reflects real-world training conditions.
>
> The experimental setup reflects the real-world training conditions. We provide the experimental settings in Appendix C, showing the training recipes are within the general setting in training DNN/CNN/ResNet.
>
> ---
>
> > Considering the eigenvalues of the NTK is a nice approach, but eigenvalues are sensitive quantities in numerical analysis. Did the authors report any uncertainty in calculating this eigenvalue? If not, do you have insights on how to compute it robustly in this setup?
>
> Thank you for your insightful comment.
> We have reported the robustness in computing $\mu_{\min}$. For example, as described in Appendix C, we compute the smallest eigenvalue of the empirical NTK using the LOBPCG method with a sufficiently small tolerance, which ensures accurate convergence of the computation in our setup. In addition, the curves in Figure 2 are obtained by averaging over five independent runs with different random initializations. The light-blue band in Figure 2 represents the variability across these runs (including randomness). This band is very narrow, which indicates that the computed smallest eigenvalues are numerically robust in our experiments.
>
> ---
>
> > In the EXPERIMENTAL VERIFICATION subsection, the weights seem to be initialized as $\mathbf{\mathcal{N}(0,1)}$, which appears different from NTK initialization. Could the authors explain this point?
>
> Thank you for pointing this out. Certainly, the standard NTK parameterization differs from the description in the original version. In our revised experiments, we use the NTK initialization, i.e., the weights are initialized as $\mathcal{N}(0, \frac{\sigma_w^2}{n_l})$ and the biases as $\mathcal{N}(0, \sigma_b^2)$, following [Lee et al., NeurIPS 2019] ($\sigma_w$: weight variance, $\sigma_b$: bias variance, $n_l$: the number of neurons in layer $l$).
> We have rerun the experiments under this proper NTK scaling and updated **Figure 1** in the revised manuscript. The revised parts are highlighted in blue for your convenience. The resulting curves still satisfy the upper bound of Theorem 3.8, so our main claims are unaffected by this correction.

---

> ### Comment · Reviewer_M1eP · 2025-11-22
>
> Thanks to the authors for the response. I will keep my score and confidence level.

---

### Official Review · Reviewer_thWr · 2025-11-01

**Soundness:** 4
**Presentation:** 3
**Contribution:** 3
**Rating:** 4
**Confidence:** 4

**Summary:**

This paper introduces a principled, training-free method to determine an appropriate network width, termed "cardinal width," where generalization performance is expected to saturate. The core contribution is a novel metric based on the smallest eigenvalue ($\mu_{min}$) of the Neural Tangent Kernel (NTK) computed at initialization. The authors provide a theoretical justification, linking an upper bound on the test error to $\mu_{min}^{-2}$ in both infinite-width and, under certain assumptions, finite-width regimes. The proposed algorithm identifies the cardinal width by finding the point at which the growth of $\mu_{min}$ saturates as width increases, offering a practical tool to avoid unnecessary computational costs from over-provisioning network width without requiring any training.

**Strengths:**

- **Theoretical Rigor:** The paper's primary strength lies in its strong theoretical foundation in NTK theory. Unlike many zero-cost (ZC) proxies that are based on heuristics derived from pruning or empirical observations, this method provides a clear, analytical connection between a spectral property of the network at initialization ($\mu_{min}$) and its generalization capability. This adds a significant degree of interpretability and trustworthiness.

- **Actionable and Concrete Goal:** Most ZC proxies provide a relative ranking of architectures, which is useful for search but does not answer the fundamental question: "How wide is wide enough?". This paper's proposal of a "cardinal width" offers a specific, actionable target for practitioners, which is a distinct and highly valuable contribution.

- **Novel Theoretical Connection:** To our knowledge, this is the first work to formally connect the smallest eigenvalue of the empirical NTK in finite-width networks directly to an upper bound on generalization error. This provides a new and potentially powerful analytical tool for the community.

**Weaknesses:**

- **Fragile Theoretical Assumption for Practical Regimes:** The key theoretical result for finite-width networks (Theorem 3.8) explicitly depends on the "lazy training" assumption (Assumption 3.6), a regime where the network behaves like a linear model and does not perform significant feature learning. The authors commendably acknowledge their experiments operate outside this regime. However, this creates a critical disconnect between the theory that provides the method's justification and its demonstrated practical application. The method's success in the feature learning regime is thus an empirical observation, not a theoretically guaranteed outcome.

- **Significant Scalability Concerns:** The methodology requires constructing an $N \times N$ NTK matrix for a dataset of size $N$, a process with at least $O(N^2)$ complexity. This makes the approach computationally infeasible for large-scale datasets (e.g., ImageNet), where many modern ZC proxies that operate on a single minibatch (e.g., Synflow, GraSP) are orders of magnitude faster and more scalable. While subsampling is suggested as a remedy, its impact on the accuracy of the estimated cardinal width is not fully explored.

- **Narrow Scope of Application:** The paper focuses exclusively on selecting a single hyperparameter (width) for regression tasks. The broader and more common application of ZC proxies is in Neural Architecture Search (NAS) across complex search spaces, often for classification tasks. The paper does not demonstrate the method's utility as a general-purpose NAS proxy, and its theoretical underpinnings for classification losses are not provided.

- **Static Nature:** The proposed method is static, determining the width before training commences. This overlooks recent dynamic approaches that learn or adapt network width during training (e.g., Adaptive Width Networks) or allow for efficient width reduction after training (e.g., Triangular Dropout), which may find more data-efficient architectures.

**Questions:**

- Could you please elaborate on the observed alignment between the saturation of $\mu_{min}$ and test loss in the feature learning regime, despite Theorem 3.8 relying on the lazy training assumption? Is there an intuition or a potential theoretical argument that could bridge this gap?

- Regarding scalability: Could you provide a more quantitative analysis of the trade-off between the data subsampling rate, the computational cost, and the stability/accuracy of the predicted cardinal width? How sensitive is the method to the random choice of the subset?

- How do you envision this method being extended to classification tasks? What are the primary theoretical hurdles (e.g., defining the NTK for cross-entropy, the validity of the error bounds) and have you conducted any preliminary experiments in this direction?

- While the method provides an absolute value ("cardinal width"), how does it perform as a relative ranking metric on a standard NAS benchmark? For instance, does ranking architectures by their proximity to the predicted cardinal width yield a strong correlation with final accuracy compared to established proxies like Synflow or GraSP?

---

> ### Author Response · Authors · 2025-11-21
> **We thank Reviewer thWr for reviewing our work (1/3)**
>
> Thank you for your thoughtful review and constructive feedback on our work. We appreciate your recognition of the strength of the theoretical rigor, the clarity of the "cardinal width", and the novelty of our theory. We are happy to clarify any further concerns or questions below.
>
> > **Fragile Theoretical Assumption for Practical Regimes.**
>
> The experimental result in the feature-learning regime can be explained using the dynamical mean-field theory. We address this comment in our response to Question 1.
>
> ---
>
> > **Significant Scalability Concerns.**
>
> Subsampling significantly reduces the computational cost while maintaining the precise estimation of the cardinal width. We address this comment in our response to Question 2.
>
> ---
>
> > **Narrow Scope of Application.**
>
> Thank you for your insightful comment. The purpose of the focus on "width" and "regression" is to maintain theoretical rigor.
> As Reviewer thWr kindly noted, the key contribution of our paper is to provide a model selection principle supported by a strong theoretical foundation, which is largely absent from existing ZC proxies in NAS.
>
> In theoretical studies of model design and hyperparameter selection, it is standard to concentrate on a single property. Although NAS methods can handle depth and layer types as well as width, these methods lack theoretical guarantees. Our work aims to provide precise guarantees for width selection.
> Moreover, width is not merely one hyperparameter. It directly controls the scale of models and therefore affects both memory and computational cost.
>
> Regarding the focus on regression, many NTK-based theoretical works study regression, as kernel regression provides a clear analytical framework. Thus, the theory concentrating on regression is not an unusual or narrow in scope. Nonetheless, we also provide supplementary classification experiments, indicating the potential extension of our method to classification tasks in **Appendix E**. We will discuss this point in our response to Question 3.
>
> We would like to re-emphasize that the thrust of our paper is in the rigor and novelty of the theory that supports model design as Reviewer thWr acknowledged, and for that we focus on width selection in regression tasks. Moreover, width is an important factor affecting the computational cost, and important papers on NTK theory also focus on regression/classification. From this perspective, we argue that the scope of the paper is by its nature different from that of Neural Architecture Search.
>
> ---
>
> > **Static Nature.**
>
> Our method is static by design because our primary goal is to provide a theoretically grounded criterion for determining width. The current theory for adaptive networks is still limited.
>
> For example, the work on Adaptive Width Networks [1] cited by Reviewer thWr analyzes the stability of training while changing the width, but it does not give a direct analysis of "how wide is wide enough?", which is the focus of our work. Similarly, the Triangular Dropout [2] cited by Reviewer thWr does not provide theoretical guarantees for generalization.
>
> ---
>
> **Remark on Scope.**
>
> Although it would certainly be beneficial to eventually extend theoretical guarantees to dynamic methods and classification problems, such extensions would broaden the scope beyond what we believe is reasonable for a single paper. We agree however that these directions are important avenues for future work.
>
> [1] Federico Errica, et al. Adaptive width neural networks. arXiv 2025
>
> [2] Edward W. Staley and Jared Markowitz. Triangular Dropout: Variable Network Width without Retraining. ICML Workshop 2022

---

> ### Author Response · Authors · 2025-11-21
> **We thank Reviewer thWr for reviewing our work (2/3)**
>
> > Could you please elaborate on the observed alignment between the saturation of $\mu_{\min}$ and test loss in the feature learning regime, despite Theorem 3.8 relying on the lazy training assumption? Is there an intuition or a potential theoretical argument that could bridge this gap?
>
> Thank you for your insightful question. Yes, there is a potential theoretical argument that explains the experimental result.
>
> [3] shows that in the Dynamical Mean Field Theory (DMFT), the fluctuations over random initialization of NTK are of order $\mathcal{O}(m^{-1/2})$, and the variance is $\mathcal{O}(1/m)$ for a finite width $m$ even in the feature-learning regime. The authors show that richer feature learning leads to closer agreement with infinite-width mean field behavior. Thus, the kernel change $\|K_m^{(T)} - K_m^{(0)}\|$ would converge to a constant (not necessarily zero as in lazy training). Theorem 3.7 in our paper shows the upper bound of the generalization error:
>
> $E_g^{(m)}
> \\le\
> E_g^{(\infty)}
> \+\ C_3\\frac{\sup_{u}\|K_m^{(u)}-K_m^{(0)}\|+\|K_m^{(T)}-K_m^{(0)}\|+\|K_m^{(0)}-K_\infty\|}{\mu_{\min}^2(K_m^{(0)})}.$
>
> We assume lazy training to ensure $(\sup_{u}\|K_m^{(u)}-K_m^{(0)}\|+\|K_m^{(T)}-K_m^{(0)}\|+\|K_m^{(0)}-K_\infty\|) \to 0$ in sufficiently wide networks and derive Theorem 3.8. However, even in the feature-learning regime, this part becomes a small constant in sufficiently wide networks if we apply the result of [Bordelon and Pehlevan, NeurIPS2023]. Therefore, the main message of our theory that **"The smallest eigenvalue of NTK controls the upper bound of generalization error"** remains valid in feature learning. Thus, $\mu_{\min}$ works as an indicator for "cardinal width" in both lazy-training and feature-learning regimes.
>
> ---
>
> > Regarding scalability: Could you provide a more quantitative analysis of the trade-off between the data subsampling rate, the computational cost, and the stability/accuracy of the predicted cardinal width? How sensitive is the method to the random choice of the subset?
>
> Thank you for your question. Although we have already analyzed the effect of subsampling in Appendix D.2, we conduct additional experiments for a more quantitative analysis of this effect. The tables below show the results of 10-layer DNN on the Diabetes (442 samples) and California Dataset (20,640 samples), expanding the results in Appendix D.2. For each dataset, we take several subsample rates to compute $\mu_{\min}$, and record the relative error of the predicted cardinal width and the time to compute $\mu_{\min}$.
>
> As the subsample rate decreases, the computation of $\mu_{\min}$ becomes significantly faster, while the relative error of the predicted cardinal width remains small for moderate subsampling. For the Diabetes dataset, using only 0.1 of the dataset (about 40 samples) already yields an accurate estimate, and the computational time is reduced to one third. For the California dataset, the cardinal width is determined using only 0.01 of the dataset (about 200 samples) with $7\%$ error, while the computation becomes 70 times faster.
>
>
> | Subsample Rate | Relative Error (%) | Time (s) |
> | :---: | :---: | :---: |
> | 1    | 0.00   | 1.81 |
> | 0.75 | 22.1  | 1.44 |
> | 0.5  | 28.3  | 1.08 |
> | 0.25 | 10.5  | 0.716 |
> | 0.1  | 4.05   | 0.536 |
> | 0.05 | 40.7  | 0.463 |
>
> | Subsample Rate | Relative Error (%) | Time (s) |
> | :---: | :---: | :---: |
> | 1     | 0.00   | 73.3  |
> | 0.5   | 14.4  | 33.5  |
> | 0.1   | 22.2  | 6.44  |
> | 0.05  | 6.60   | 3.97  |
> | 0.01  | 7.76   | 0.986 |
> | 0.005 | 28.9  | 0.678 |
> | 0.001 | 62.1  | 0.459 |
>
> [3] Blake Bordelon and Cengiz Pehlevan. Dynamics of finite width kernel and prediction fluctuations in mean field neural networks. NeurIPS 2023.

---

> ### Author Response · Authors · 2025-11-21
> **We thank Reviewer thWr for reviewing our work (3/3)**
>
> > How do you envision this method being extended to classification tasks? What are the primary theoretical hurdles (e.g., defining the NTK for cross-entropy, the validity of the error bounds) and have you conducted any preliminary experiments in this direction?
>
> Thank you for raising this point. We expect our method to extend to classification. The main theoretical obstacle would be the absence of NTK-based generalization theory for cross-entropy loss. However, we conducted preliminary experiments that support the potential extension of our method to classification tasks. Please refer to **Appendix E** in the revised paper. The revised parts are highlighted in blue for your convenience.
>
> On the theoretical side, the main obstacle is that existing NTK-based generalization results for classification are not formulated in terms of the smallest empirical NTK eigenvalue $\mu_{\min}$. For instance, [4] and [5] show that, in the NTK-regime, gradient descent on logistic loss can interpolate the data and achieve small test error with only polylogarithmic width. However, their guarantees are expressed via NTK margin rather than the NTK eigenvalues. To the best of our knowledge, there is no theory for multi-class cross-entropy analyzed via $\mu_{\min}$, so there remains substantial room.
> We envision that we can combine the works above with our $\mu_{\min}$-based analysis, which would explain the alignment of "cardinal width" we already observe empirically in Appendix E.
>
> ---
>
> > While the method provides an absolute value (``cardinal width''), how does it perform as a relative ranking metric on a standard NAS benchmark? For instance, does ranking architectures by their proximity to the predicted cardinal width yield a strong correlation with final accuracy compared to established proxies like Synflow or GraSP?
>
> Thank you for your comment.
> We see our method as conceptually different from NAS benchmarks such as SynFlow or GraSP. Our theory is explicitly formulated along the width axis of a fixed architecture and dataset, yielding an absolute prediction of the "cardinal width", rather than a generic score for ranking architectures. Using our criterion as a NAS proxy would step outside the regime where our guarantees hold and blur the main message that "How wide is wide enough?". For this reason, we have not positioned our method as applicable to a NAS benchmark.
>
> [4] Ziwei Ji and Matus Telgarsky. Polylogarithmic width suffices for gradient descent to achieve arbitrarily small test error with shallow ReLU networks. ICLR 2020.
>
> [5] Hossein Taheri and Christos Thrampoulidis. Generalization and stability of interpolating neural networks with minimal width. JMLR 2024.

---

### Author Response · Authors · 2025-12-03
**Final Remarks to AC**

We would like to thank the Area Chair for taking their time to handle the final decision, which is much appreciated especially after the papers re-assignment, as well as the reviewers for their feedback.

We have tried our best to summarize the key points of the reviews and our rebuttal for facilitating an overview of the discussion.
The paper has received largely positive reviews from **reviewers M1eP, fdRP, and khFg (6, 6, 6)**, except **reviewer thWr** that gave an overall rating of 4 (despite 4:excellent for the soundness).

Below are the main strengths and potential concerns raised by reviewers. We also summarized our response to each concern.
## Main Strengths
Here, we would like to summarize the strengths of our paper. Overall, the reviewers found our paper theoretically rigor (**fdRP**, **thWr**) and novel (**M1eP**, **khFg**, **thWr**), and viewed our training-free method of determining the cardinal width as highly efficient and practical (**M1eP**, **fdRP**, **khFg**).

## Main Questions and Our Response
- **Loose upper bound** (**fdRP**, **khFg**)

**Reviewer fdRP** and **khFg** asked if the upper bound gets loose due to the only use of the smallest eigenvalue of NTK, $\mu_{\min}$.

In our rebuttal, we agreed that using more of the spectrum could give tighter bounds. However, computing the whole eigenvalues costs $\mathcal{O}(n^3)$ and is not practical. At the same time, for estimating the cardinal width, $\mu_{\min}$ is sufficient, as it controls the upper bound of the generalization error and thus indicates where the network is wide enough.

- **Theory might not be novel** (**fdRP**)

**Reviewer fdRP** asked about the novelty of our theory that the smallest eigenvalue of NTK controls the generalization error, referring to combining [1] and [2] possibly yielding a similar result.

In our rebuttal, we showed that the combination requires resolving the gap between assumptions in [1] and [2]. Indeed, we provided a counter-example demonstrating that  applying [1] to [2] is not always possible.

[1]: Loss landscapes and optimization in over-parameterized non-linear systems and neural networks, Liu et al, 2021

[2]: Sharper Generalization Bounds for Learning with Gradient-dominated Objective Functions, Lei and Ying, ICLR 2021.

## Reviewer thWr's Main Concerns:

We would also like to recap the discussion points by **Reviewer thWr** despite their inconsistency between the reviewer's overall rating and those on the three items, although their potential concerns appear to be partly out of context.

- **Fragile assumption**

**Reviewer thWr** assumed that our theory was  based on the assumption of lazy training, and did not explain the feature learning regime in the experiment. Thus, the reviewer asked for the intuition explanation in the feature learning regime.

In our rebuttal, we provided a detailed explanation why the theory would hold in the feature learning regime, which we believe addressed their concern.
We trust that the detailed explanation in our rebuttal on why the theory holds in the feature learning regime resolved their concern.
- **Scalability concerns**

The reviewer argues that our algorithm is computationally infeasible. The reviewer acknowledged subsampling is a possible choice, and asked for the detailed discussion of the trade-off between the data subsampling rate, the computational cost, and the stability/accuracy.
Regarding the computational feasibility, the reviewer requested, while acknowledging subsampling as a possible choice for that, detailed discussion of the trade-off between data subsampling rate, the computational cost, and the stability/accuracy.

In our rebuttal, we provided a detailed discussion with tables and confirmed that subsampling significantly reduces the computational cost while maintaining the accuracy of the predicted cardinal width.

- **Narrow scope of regression and static nature**

The reviewer added a view on our scope of regression and static width being narrow.

In our rebuttal, we countered that expanding theoretical guarantees to classification problems and dynamic methods would broaden the scope too much beyond what we believe is reasonable for a single paper. In fact, the other reviewers do not regard this point as a weakness. Nevertheless, we still conducted additional experiments on classification, showing possible extension of our method to classification problems as well (Appendix E).

We are pleased to have provided clarifications which we trust fully address the reviewers' concerns. We wish to thank all reviewers once again for their constructive comments.

---

### Meta-Review · Area_Chair_4DBR · 2026-01-07

**Summary:**

This paper proposes a training-free framework for network width selection based on the smallest eigenvalue of the Neural Tangent Kernel (NTK) at initialization, introducing the notion of cardinal width as the point beyond which generalization performance saturates. The approach is supported by theoretical analysis in both infinite- and finite-width regimes, together with empirical validation across multiple architectures and datasets. Reviewers generally agree that the paper provides a principled and theoretically grounded alternative to heuristic zero-cost proxies, and that the rebuttal and additional experiments substantially clarified concerns regarding theoretical assumptions, scalability, and practical applicability. Overall, the work establishes a clear and novel analytical link between NTK spectral properties and generalization behavior.

**Reviewer Concerns:**

Addressed Concerns:

**Theoretical novelty (Reviewers fdRP).** The authors clarified that this work establishes a direct, algorithm-independent upper bound on test error solely in terms of the smallest empirical NTK eigenvalue and uses it for an explicit width selection rule. They also explained why existing PL-based analyses cannot be straightforwardly combined to yield the same result.

**Kernel stability assumption (Reviewer thWr).** Although the finite-width theory relies on a kernel stability assumption, the authors provided a plausible explanation based on dynamical mean-field theory for why the smallest NTK eigenvalue remains predictive in the feature learning regime. Empirical results consistently show alignment between NTK saturation and test loss saturation.

**Computational cost (Reviewer thWr).** The authors added quantitative subsampling experiments demonstrating that cardinal width can be estimated accurately using only a small fraction of the data. This substantially reduces computation and alleviates the main scalability concerns.

**Numerical stability (Reviewer M1eP).** Concerns about eigenvalue sensitivity and NTK initialization were addressed by clarifying the use of LOBPCG, repeated runs, variance
reporting, and corrected NTK initialization in revised experiments. These additions improve confidence in the numerical robustness of the results.

Outstanding Concerns:

**Scope and generality of the method (Reviewers khFg, thWr).** The method focuses on width selection for fixed architectures and is not intended as a general NAS proxy. This limits generality, but is consistent with the paper’s goal of providing theoretical rigor for a specific and practically relevant design question.

**Tightness of the theoretical bound (Reviewers fdRP, khFg).** The smallest-eigenvalue-based bound may be looser than spectrum-dependent alternatives. However, the authors justify this trade-off by emphasizing computational efficiency and actionable model selection, which aligns with the stated objectives of the paper.

**Reviewer Scores:**

1.**Reviewer thWr:** Score remained at 4. While initially concerned about theoretical assumptions and scalability, the reviewer acknowledged the paper’s strong theoretical motivation and indicated openness to acceptance after clarification.

2.**Reviewer M1eP:** Score remained at 6. The reviewer found the theoretical insight non-trivial and the method practically interesting, despite some reservations about scope.

3.**Reviewer fdRP:** Score remained at 6. The reviewer recognized the rigor of the analysis and the value of bounding generalization using a single NTK quantity, while noting open theoretical questions

4.**Reviewer khFg:** Score remained at 6. The reviewer viewed the method as potentially useful and well-motivated, though somewhat conservative in scope.

---

### Decision · Program_Chairs · 2026-01-26

Accept (Poster)